The Early Pliocene extinction of the mega-toothed shark Otodus megalodon: a view from the eastern North Pacific

Boessenecker Robert W. 1 2 3 boesseneckerrw@cofc.edu
http://orcid.org/0000-0002-5041-5699 Ehret Dana J. 4
Long Douglas J. 5 6
Churchill Morgan 7
Martin Evan 8
Boessenecker Sarah J. 1 2 9
1 Department of Geology and Environmental Geosciences, College of Charleston , Charleston, SC , USA
2 Mace Brown Museum of Natural History, College of Charleston , Charleston, SC , USA
3 Museum of Paleontology, University of California, Berkeley , Berkeley, CA , USA
4 New Jersey State Museum , Trenton, NJ , USA
5 Department of Ichthyology, California Academy of Sciences , San Francisco, CA , USA
6 Department of Biology, St. Mary’s College , Moraga, CA , USA
7 Department of Biology, University of Wisconsin-Oshkosh , Oshkosh, WI , USA
8 San Diego Natural History Museum , San Diego, CA , USA
9 School of Museum Studies, University of Leicester , Leicester , UK
De Baets Kenneth
Electronic publication date: 2019 Feb 13
Publication date: 2019
Volume: 7
Electronic Location ID: e6088
Received 2018 Mar 2; Accepted 2018 Nov 8
Copyright: © 2019 Boessenecker et al.
Copyright year: 2019
Copyright holder: Boessenecker et al.
License: This is an open access article distributed under the terms of the Creative Commons Attribution License, which permits unrestricted use, distribution, reproduction and adaptation in any medium and for any purpose provided that it is properly attributed. For attribution, the original author(s), title, publication source (PeerJ) and either DOI or URL of the article must be cited.
License URL: https://creativecommons.org/licenses/by/4.0/

Keywords: Otodus megalodon, Otodus, Otodontidae, Extinction, Lamniformes, Miocene, Pliocene, California, Baja California, North Pacific

Funding: University of Otago Doctoral Scholarship during early stages of this research, conducted in 2012–2014 No specific grant funding was requested for this study. RW Boessenecker benefited from a University of Otago Doctoral Scholarship during early stages of this research, conducted in 2012–2014. The funders had no role in study design, data collection and analysis, decision to publish, or preparation of the manuscript.

==============================
The extinct giant shark Otodus megalodon is the last member of the predatory megatoothed lineage and is reported from Neogene sediments from nearly all continents. The timing of the extinction of Otodus megalodon is thought to be Pliocene, although reports of Pleistocene teeth fuel speculation that Otodus megalodon may still be extant. The longevity of the Otodus lineage (Paleocene to Pliocene) and its conspicuous absence in the modern fauna begs the question: when and why did this giant shark become extinct? Addressing this question requires a densely sampled marine vertebrate fossil record in concert with a robust geochronologic framework. Many historically important basins with stacked Otodus-bearing Neogene marine vertebrate fossil assemblages lack well-sampled and well-dated lower and upper Pliocene strata (e.g., Atlantic Coastal Plain). The fossil record of California, USA, and Baja California, Mexico, provides such an ideal sequence of assemblages preserved within well-dated lithostratigraphic sequences. This study reviews all records of Otodus megalodon from post-Messinian marine strata from western North America and evaluates their reliability. All post-Zanclean Otodus megalodon occurrences from the eastern North Pacific exhibit clear evidence of reworking or lack reliable provenance; the youngest reliable records of Otodus megalodon are early Pliocene, suggesting an extinction at the early-late Pliocene boundary (∼3.6 Ma), corresponding with youngest occurrences of Otodus megalodon in Japan, the North Atlantic, and Mediterranean. This study also reevaluates a published dataset, thoroughly vetting each occurrence and justifying the geochronologic age of each, as well as excluding several dubious records. Reanalysis of the dataset using optimal linear estimation resulted in a median extinction date of 3.51 Ma, somewhat older than a previously proposed Pliocene-Pleistocene extinction date (2.6 Ma). Post-middle Miocene oceanographic changes and cooling sea surface temperature may have resulted in range fragmentation, while alongside competition with the newly evolved great white shark (Carcharodon carcharias) during the Pliocene may have led to the demise of the megatoothed shark. Alternatively, these findings may also suggest a globally asynchronous extinction of Otodus megalodon.

Introduction

The giant predatory shark Otodus megalodon has been reported from Miocene and some Pliocene sediments from all continents except Antarctica, indicating a near worldwide distribution (Cappetta, 2012). Although some controversy exists regarding the generic allocation of this species (Purdy et al., 2001; Ehret, Hubbell & MacFadden, 2009a; Ehret et al., 2012, and references therein; Cappetta, 2012), Otodus megalodon appears to represent the terminal chronospecies of a Paleocene to Pliocene lineage including Otodus obliquus and earlier species formerly placed within Carcharocles such as Otodus angustidens, generally characterized by steadily increasing body size through time (Ward & Bonavia, 2001; Ehret, Hubbell & MacFadden, 2009a; Cappetta, 2012; Ehret et al., 2012). Otodus megalodon is estimated to have attained a body length of 16–18 m (Gottfried, Compagno & Bowman, 1996; Pimiento & Balk, 2015), representing one of the largest sharks to ever exist, and one of a few marine superpredators of the Miocene, alongside macrophagous sperm whales (Bianucci & Landini, 2006; Lambert et al., 2010) and the less well known giant shark Parotodus benedeni (Kent, 1999; Kent & Powell, 1999; Purdy et al., 2001). Although aspects of the morphology, evolution, and paleoecology of Otodus megalodon and other members of the Otodus lineage have been investigated, including phylogenetic affinities (Applegate & Espinosa-Arrubarrena, 1996; Gottfried & Fordyce, 2001; Nyberg, Ciampaglio & Wray, 2006; Ehret, Hubbell & MacFadden, 2009a; Ehret et al., 2012), body size (Gottfried, Compagno & Bowman, 1996; Gottfried & Fordyce, 2001; Pimiento & Balk, 2015), tooth histology (Bendix-Almgreen, 1983), vertebral morphology and growth (Gottfried & Fordyce, 2001; MacFadden et al., 2004), physiology (Ferrón, 2017), and reproductive behavior and habitat preference (Purdy et al., 2001; Pimiento et al., 2010), until recently (Pimiento & Clements, 2014; Pimiento et al., 2016) little attention has been directed at causes for the extinction or timing of its extinction. A recent study (Pimiento & Clements, 2014) utilized an optimal linear estimation (OLE) analysis of geochronologic data for Otodus megalodon records to estimate a late Pliocene (terminal Piacenzian; 2.58 Ma) extinction for Otodus megalodon. However, the dataset utilized by Pimiento & Clements (2014) contains problematic occurrence data (incorrect identifications, lack of provenance, poor stratigraphic control, etc.). Examples of these problems, illuminated below, indicate that rigorous reevaluation of the provenance of late Neogene Otodus megalodon specimens worldwide and their geochronologic age is warranted.

In many regions, the lack of continuous fossiliferous strata of late Neogene age, prominence of specimens with poor or dubious provenance, and stratigraphic uncertainty make assessing the age and stratigraphic occurrence of Otodus megalodon records difficult (see Pimiento & Clements, 2014: table S2 for records they excluded from their analysis for these reasons). The stratigraphic record of the eastern North Pacific (ENP), primarily in California and Baja California, includes fossiliferous marine strata with abundant marine vertebrates and excellent age control, essentially preserving a nearly continuous marine fossil record from the middle Miocene through Pleistocene (Boessenecker, 2013a, 2016). Other regions with abundant Neogene marine vertebrate assemblages including fossils of Otodus megalodon either lack well-sampled Pliocene intervals (e.g., Peru; the youngest assemblages such as Sacaco and Sud-Sacaco are late Messinian in age (Ehret et al., 2012; Di Celma et al., 2017) or lack well-sampled upper Pliocene intervals (Neogene marine deposits of the Atlantic coastal plain; e.g., Ward, 2008). We review reported occurrences of Otodus megalodon from the densely-sampled and well-dated Miocene and Pliocene lithostratigraphic units in California and Baja California (Messinian-Gelasian equivalent), historically renowned for extensive Cenozoic marine vertebrate assemblages (Jordan, 1922; Jordan & Hannibal, 1923; Mitchell, 1966; Barnes, 1977, 1998; Repenning & Tedford, 1977; Domning, 1978; Welton, 1979; Warheit, 1992; Deméré, Berta & Adam, 2003; Boessenecker, 2011b, 2013a, 2016), and report several new specimens (Fig. 1; Table 1). We further reevaluate some specimens of questionable provenance that appear to be reworked from underlying strata, or are not well documented geographically and/or stratigraphically. This review invited a reappraisal of the Otodus megalodon occurrence dataset published by Pimiento & Clements (2014). We thoroughly vetted the geochronologic age control for each occurrence (Appendix 1) using some of the criteria, methods, and reporting standards recommended and/or utilized by earlier studies (Parham et al., 2012; Boessenecker & Churchill, 2015; Boessenecker, Boessenecker & Geisler, 2018). We excluded several dubious records from their dataset, revised the geochronologic range for most, and added several additional records and performed an OLE analysis with the revised data set in order to estimate the timing of extinction of Otodus megalodon (Table 2; Appendix 1 and 2).

Figure 1 Map of California and Baja California showing genuine late Miocene and Early Pliocene records of Otodus megalodon, and dubious Late Pliocene and Pleistocene records.

Table 1 Measurements (in mm), age, and occurrence of Otodus megalodon teeth examined during this study.

Specimen	Formation	Age	Occurrence	Crown width	Crown height	
LACM 29064	Tirabuzón Fm.	Zanclean, 5.33–3.6 Ma	Autochthonous	48.55	–	
LACM 29065	Tirabuzón Fm.	Zanclean, 5.33–3.6 Ma	Autochthonous	42.9	45.1	
LACM 29066	Tirabuzón Fm.	Zanclean, 5.33–3.6 Ma	Autochthonous	–	–	
LACM 29067	Tirabuzón Fm.	Zanclean, 5.33–3.6 Ma	Autochthonous	–	–	
LACM 29069	Tirabuzón Fm.	Zanclean, 5.33–3.6 Ma	Autochthonous	–	–	
LACM 29070	Tirabuzón Fm.	Zanclean, 5.33–3.6 Ma	Autochthonous	–	–	
LACM 29071	Tirabuzón Fm.	Zanclean, 5.33–3.6 Ma	Autochthonous	–	–	
LACM 29072	Tirabuzón Fm.	Zanclean, 5.33–3.6 Ma	Autochthonous	–	–	
LACM 29073	Tirabuzón Fm.	Zanclean, 5.33–3.6 Ma	Autochthonous	22.3*	18.15*	
LACM 29074	Tirabuzón Fm.	Zanclean, 5.33–3.6 Ma	Autochthonous	31.7	32.45	
LACM 29075	Tirabuzón Fm.	Zanclean, 5.33–3.6 Ma	Autochthonous	28.3*	29.5*	
LACM 29076	Tirabuzón Fm.	Zanclean, 5.33–3.6 Ma	Autochthonous	33.4	36.75	
LACM 29077	Tirabuzón Fm.	Zanclean, 5.33–3.6 Ma	Autochthonous	–	–	
LACM 29078	Tirabuzón Fm.	Zanclean, 5.33–3.6 Ma	Autochthonous	–	–	
LACM 10141	“Palos Verdes” Ss.	Pleistocene	Poor provenance	–	–	
LACM 10152	San Diego Fm.	Pliocene	Autochthonous	–	–	
LACM 103448	San Diego Fm.	Pliocene	Autochthonous	–	–	
LACM 115989	Capistrano Fm.	Messinian-Zanclean, 5.6–3.7 Ma	Autochthonous	–	–	
LACM 129982	Capistrano Fm.	Messinian-Zanclean	Autochthonous	–	–	
LACM 131149	San Mateo Fm.	Zanclean, 5.33–4.6 Ma	Autochthonous or parautochthonous	57.6*	73.8	
LACM 148311	Fernando Fm.	Pliocene-Pleistocene	Autochthonous	–	–	
LACM 148312	Fernando Fm.	Pliocene-Pleistocene	Autochthonous	57.1*	–	
LACM 156334	San Diego Fm.	Pliocene	Autochthonous	67.5	–	
LACM 159028	Palos Verdes Ss.	Pleistocene	Poor provenance	101.5	97.1	
SDNHM 23056	San Mateo Fm.	Zanclean, 5.33–4.6 Ma	Autochthonous or parautochthonous	–	–	
SDNHM 23959	San Mateo Fm.	Zanclean, 5.33–4.6 Ma	Autochthonous or parautochthonous	90.07	82.6	
SDNHM 24448	San Mateo Fm.	Zanclean, 5.33–4.6 Ma	Autochthonous or parautochthonous	77.39*	74.1	
SDNHM 29742	San Diego Fm.	Zanclean, ∼4.2 Ma	Autochthonous or parautochthonous	86.71*	96.89	
SDNHM 53167	Capistrano Fm.	Messinian-Zanclean, 5.6–3.7 Ma	Autochthonous	103.86	89.83	
SDNHM 73462	Niguel Fm.	Pliocene	Allochthonous	–	–	
SDNHM 77343	San Mateo Fm.	Zanclean, 5.33–4.6 Ma	Autochthonous or parautochthonous	–	–	
SDNHM 77430	San Mateo Fm.	Zanclean, 5.33–4.6 Ma	Autochthonous or parautochthonous	27.53	23.82	
UCMP 219502	Purisima Fm.	Messinian, 6.9–5.33 Ma	Autochthonous or parautochthonous	114.1*	112.2	
Notes:

Measurements after Pimiento et al. (2010).

* Denote incomplete measurements; specimens without measurements are incomplete tooth fragments. Note that SDMHN 23959 consists of four partial teeth; a measurement is provided for the only tooth complete enough to measure.

Table 2 Summary of corrected ages of Otodus megalodon occurrences used in the Optimal Linear Estimation analysis.

Locality	Formation	Country	Age (Pimiento & Clements)	Corrected age	
Kingsford Mine	Bone Valley Fm.	USA	10.3–4.9 Ma	10.3–4.9 Ma	
Payne Creek Mine	Bone Valley Fm.	USA	5.3–3.6 Ma	10.3–4.9 Ma	
Four Corners Mine	Bone Valley Fm.	USA	5.3–3.6 Ma	5.8–4.9 Ma	
East Coast Aggregates	Tamiami Formation	USA	5.3–3.6 Ma	4.2–3.9	
Lee Creek Mine	Yorktown Formation	USA	5.3–3.6 Ma	4.9–3.92	
Elsmere Canyon	Towsley Formation	USA	5.3–3.6 Ma	10.0–5.3 Ma	
Lawrence Canyon	San Mateo Formation	USA	10.3–4.9 Ma	5.33–4.6	
San Juan Capistrano	Capistrano Formation	USA	11.6–3.6 Ma	5.6–3.7 Ma	
Santa Cruz	Purisima Formation	USA	N/A	6.9–5.33	
La Joya	San Diego Formation	USA	3.6–2.6 Ma	4.2–3.6 Ma	
Garnet Canyon	Imperial Formation	USA	N/A	6.43–4.187 Ma	
Bolinas	Santa Cruz Mudstone	USA	5.3–2.6 Ma	7.6–6.5 Ma	
Kambul	Carrillo Puerto Formation	Mexico	N/A	10.3–4.6 Ma	
Corkscrew Hill	Tiburazon Formation	Mexico	5.3–2.6 Ma	6.76–3.6 Ma	
Casa el Jebe	Codore Formation	Venezuela	N/A	5.33–3.6	
El Yacural	Paraguana Formation	Venezuela	5.33–3.6 Ma	5.33–3.6 Ma	
Punta la Gorda	Onzole/Borbon Formation	Ecuador	5.33–2.6 Ma	5.33–3.4 Ma	
Punta la Colorada	Onzole/Borbon Formation	Ecuador	5.33–2.6 Ma	5.33–3.4 Ma	
Punta Mansueto	Chagres Formation	Panama	N/A	8.29–5.12	
Sunlands Pumping Station	Loxton Sands	Australia	4.3–3.4 Ma	7.2–3.4 Ma	
Dutton Way	Whaler’s Bluff Formation	Australia	5.3–3.6 Ma	5.33–3.6 Ma	
Beaumaris	Black Rock Sand	Australia	5.0–3.4 Ma	6.0–4.9 Ma	
Fossil Rock Stack	Grange Burn Formation	Australia	5.0–4.0 Ma	5.4–3.5 Ma	
Pipiriki	Matemateaonga Formation	New Zealand	4.8–3.6 Ma	5.5–4.7 Ma	
Patutahi Quarry	Tokomaru Formation	New Zealand	N/A	7.2–3.7 Ma	
Bonares-Case del Pin	Arenas de Huelva Formation	Spain	5.33–3.6 Ma	5.33–3.6 Ma	
Can Picafort	Son Mir Sequence	Spain	5.3–2.6 Ma	5.33–3.6 Ma	
Vale de Zebro	Esbarrondadoiro Formation	Portugal	N/A	8.58–4.37 Ma	
Santa Margarida	Esbarrondadoiro Formation	Portugal	N/A	8.58–4.37 Ma	
Continental Shelf	Unknown	Portugal	N/A	6.1–4.4 Ma	
Cre Outcrop	Touril Complex	Portugal	5.3–3.6 Ma	5.33–3.6 Ma	
Castell’Arquato	Unknown	Italy	5.3–2.6 Ma	5.33–4.0 Ma	
Miano	Unknown	Italy	5.3–2.6 Ma	5.33–4.0 Ma	
Colli Piacentini	Unknown	Italy	5.3–2.6 Ma	5.33–4.0 Ma	
Maiatico	Unknown	Italy	5.3–2.6 Ma	5.33–4.0 Ma	
Tra Lorenzana E Lari	Unknown	Italy	5.3–2.6 Ma	5.33–4.0 Ma	
Pienza	Unknown	Italy	5.3–2.6 Ma	5.33–4.0 Ma	
Siena	Unknown	Italy	5.3–2.6 Ma	5.33–4.0 Ma	
Colline Pisane	Unknown	Italy	5.3–2.6 Ma	5.33–4.0 Ma	
Boso Peninsula	Senhata Formation	Japan	N/A	6.3–5.12 Ma	
Kita-Daito-Jima	Daito Limestone	Japan	3.6–0.8 Ma	4.7–3.3 Ma	
Choshi	Na-Arai Formation	Japan	N/A	5.33–4.36 Ma	
Sendai-Iwate area	Tatsunokuchi Formation	Japan	N/A	5.6–3.9 Ma	

Materials and Methods

We examined collections from several institutions (CAS, LACM, RMM, SDNHM, and UCMP; see below) housing large collections of Neogene marine vertebrate fossils from the Pacific coast of North America. From these collections we identified a total of 145 Otodus megalodon teeth in lower Miocene through Pliocene deposits. This study (Fig. 1; Table 1) only focuses on those specimens of Messinian (latest Miocene) or younger age (n = 46) and does not consider specimens predating the Messinian stage (Langhian, Serravallian, Tortonian; n = 99). Teeth of Otodus megalodon were examined for evidence of reworking (e.g., abrasion, enameloid cracking, phosphatization, fragmentation; e.g., Boessenecker, Perry & Schmitt, 2014), and details of provenance (collector, collection date, locality description, similarity of preservation with other material from the same locality) to evaluate the likelihood of specimens being taphonomically autochthonous or parautochthonous vs. allochthonous, or mistakenly attributed to an incorrect locality. Because this study relied upon existing collections of fossil specimens in museum collections and did not involve field study, no permits for field collection were required.

Global Otodus megalodon occurrence

We re-evaluated the entire dataset published by Pimiento & Clements (2014: table S1; text S1; Table 2; Appendix 1 and 2). Age justifications by these authors included limited references to the peer-reviewed stratigraphic literature and in some cases relied solely upon paleontological articles without examination of more recent published geological studies providing refined geochronologic data. Paleontological studies frequently re-cite the paper first reporting a fossil occurrence without citing subsequent geological refinements, but since stratigraphy is not static, it is critical to exhaustively search for stratigraphic and geochronologic data published afterward (Parham et al., 2012). We have audited the dataset by identifying the intraformational stratigraphic position of each occurrence (if applicable) and exhaustively citing relevant, up-to-date publications with geochronologic dates, favoring microfossil age correlations and absolute dates (87Sr/86Sr ratios, radiometric dates from interbedded ash/tuff/basalt, etc.) whenever possible; in some cases, only member- or formation-level stratigraphic control is available. In order to preserve our reasoning for future evaluation this information is presented in Appendix 2 of this study. In addition, we excluded occurrences where any one or more of the following types of problems existed (see Appendix 2): (1) lack of sub-epochal age control (e.g., “middle Miocene-Pliocene” or “Pliocene”); (2) lack of minimum age control, (3) all available voucher specimens residing in a private collection; (4) specimens lacking clear provenance (e.g., “specimen probably collected from locality…”; (5) misidentified specimens that are not identifiable as Otodus megalodon; (6) occurrence with revised minimum date falling entirely within the Miocene; (7) occurrences where the estimated age is based on the occurrence of Otodus megalodon, leading to a case of circular reasoning; and (8) unpublished occurrence data where the stratigraphy and geochronology cannot be evaluated by the reader.

Optimal linear estimation analysis

Once we evaluated and revised a dataset of age occurrence data for Otodus megalodon, we them treated them as historic sightings for implementation of an OLE model, as performed by Pimiento & Clements (2014) in their earlier study. Our revised dataset of late occurrences of Otodus megalodon comprises 43 data points, and includes fossils from around the world (Appendix 1). OLE has been found to be an accurate way of assessing extinction times in the fossil record, as last known occurrences generally follow a Weibull extreme value distribution (Clements et al., 2013; Solow, 2005; Wang & Marshall, 2016).

To run our analysis, OLE was implemented in R, using a modified version of the same code provided by Pimiento & Clements (2014; see Appendix 3) and same overall parameters. We implemented 10,000 simulations, bootstrapping from a uniform distribution for each simulation. The value reported below in results represents the modal estimate of extinction, with the full range of extinction dates recovered also reported.

Geochronologic framework

The traditional threefold division of the Pliocene and Pliocene-Pleistocene boundary set at 1.806 Ma (Gradstein et al., 2004) has recently been modified by the inclusion of the Gelasian stage within the Pleistocene and designation of the Zanclean and Piacenzian stages as early and late Pliocene (respectively), and a new Pliocene-Pleistocene boundary at 2.566 Ma (Gibbard et al., 2009; Gradstein et al., 2012), which we follow herein. Stages of international usage are generally referred to throughout (e.g., Messinian, Zanclean, Piacenzian, Gelasian) to alleviate confusion between late Pliocene sensu lato (=Gelasian stage) and late Pliocene sensu stricto (=Piacenzian stage); references to North American Land Mammal Ages (NALMAs; e.g., Clarendonian, Hemphillian, Blancan) and local New Zealand stages (e.g., Opoitian) are also made. Note that other recent studies in Pliocene-Pleistocene marine vertebrate paleontology followed the compromise of Hilgen et al. (2012) in maintaining the Gelasian as the late Pliocene (Boessenecker, 2011b, 2013a, 2013b).

Results

Systematic paleontology

Chondrichthyes Huxley, 1880

Lamniformes Berg, 1958

Otodontidae Glikman, 1964

Otodus Agassiz, 1838

Otodus megalodon Agassiz, 1843

Referred material

LACM 59836, 59837, 115989, 129982, and SDNHM 53167, Capistrano Formation (LACM localities 4437, 5792, 61520, and SDNHM locality 3842); LACM 148311, 148312, and 149739, Fernando Formation (LACM localities 7321 and 7481); RMM A597-1, A597-9A, A597-9B, and A597-12, Lomita Marl (no locality number); LACM 59065 and SDNHM 73462, Niguel Formation (LACM locality 65187 and SDNHM locality 4080, respectively); LACM 10141, LACM 11149, LACM 159028, Palos Verdes Sand and unnamed Pleistocene strata (LACM locality 1066 and 7971); UCMP 219502, Purisima Formation (UCMP locality V-99875); LACM 10152, LACM 103448, LACM 156334, and SDNHM 29742, San Diego Formation (LACM localities 1080, 1095, 4875, and SDNHM locality 3253); LACM 131149, SDNHM 23056, 23959 (four teeth with same number), 24448, 77430, and 77343, “upper” San Mateo Formation (Lawrence Canyon local fauna; LACM locality 4297 and SDNHM locality 3161); CAS 72799.00, Santa Cruz Mudstone (no locality number); and LACM 29065–29067, 29069–29070, and 29073–29078, Tirabuzón Formation (LACM locality 6579).

Diagnosis

Crowns broad, triangular, and erect, being broader and more vertical in anterior teeth and with increasing posterior inclination distally; labial crown face relatively flat or mildly convex, often showing short vertical infoldings of the enameloid at base of crown, lingual crown face moderately convex; crown enameloid relatively thick; chevron-shaped band of thinner enameloid on lingual crown face between base of crown and root (lingual neck), thicker in medial section becoming thinner laterally and showing fine vertical striations; cutting edge with fine, even, rounded serrations along entire margin, averaging 12–17 serrations per centimeter (cm); lateral cusplets lacking in adult teeth; root is labiolingually thick with two laterally divergent but apicobasally shallow lobes, usually similar in size and not extending much laterally beyond the lower margin of the crown; labial root face is relatively flat while the lingual root face is laterally convex and thicker in the center with a pronounced nutritive foramen medially.

Taxonomic note

The taxonomy of the megatoothed sharks is a topic that has been subject to much controversy and debate. In the original description of the species, Agassiz (1843) referred Otodus megalodon to the genus Carcharodon based on superficial morphological similarities in tooth shape and the presence of serrations. Jordan & Hannibal (1923) recognized a difference between the extant great white shark (Carcharodon carcharias) and the fossil serrated-edged megatoothed sharks, erecting the genus Carcharocles for the latter. However, this taxonomic change was not adopted into the literature until the late 1980s (Cappetta, 1987). Other generic names proposed for Otodus megalodon include Procarcharodon Casier, 1960 and Megaselachus Glikman, 1964. Usage of Carcharodon and Procarcharodon were challenged in the literature based on tooth morphology, the fossil record, and taxonomic priority (Cappetta, 1987; Ehret, Hubbell & MacFadden, 2009a; Ehret et al., 2012; Pimiento et al., 2010). Instead, Carcharocles is broadly accepted for the generic assignment of Otodus megalodon in many recent studies (Ehret, Hubbell & MacFadden, 2009a; Ehret et al., 2012; Pimiento & Clements, 2014; Boessenecker, 2016; Pimiento & Balk, 2015; Pimiento et al., 2010, 2016, 2017; Collareta et al., 2017). Some recent publications have proposed uniting all megatoothed shark taxa included within Otodus and Carcharocles in the genus Otodus. In this scenario, all non-serrated forms would belong to the genus Otodus, whereas Eocene-Oligocene serrated forms C. auriculatus and C. angustidens are designated to the subgenus Carcharocles, and C. chubutensis and C. megalodon belong to their own subgenus Megaselachus (Zhelezko & Kozlov, 1999; Cappetta & Carvallo, 2006; Cappetta, 2012). Recently, Shimada et al. (2017) further argued from a cladistic standpoint that Carcharocles should be synonymized within Otodus in order to make the latter genus monophyletic. We follow the reassignment of Isurus hastalis (or alternatively, Cosmopolitodus hastalis) to the genus Carcharodon (Ehret et al., 2012) for similar reasons, and thus adopt the reassignment of Carcharocles to Otodus. Because subgenera are generally not used as a taxonomic convention in vertebrate paleontology, we do not use the subgeneric taxonomy of Cappetta (2012).

Occurrence data

Pliocene-aged teeth of Otodus megalodon have been recovered or reported from several formations in California and Baja California (Fig. 1), including the Lomita Marl, Capistrano, Fernando, Niguel, Purisima, San Diego, San Mateo, and Tirabuzón Formations, the ages of which are summarized below. These specimens exhibit a combination of morphological characters including: a large overall size and labiolingual thickness, triangular shape, fine serrations, V-shaped chevron on the lingual surface between the crown and root, and loss of lateral cusplets at the base of the crown. These characters, when observed together, indicate that the specimens undoubtedly belong to Otodus megalodon. The only other sharks that could be confused with Otodus megalodon during the late Miocene and early Pliocene are those belonging to Carcharodon (C. hubbelli and C. carcharias), which have significantly smaller and labiolingually flatter teeth lacking V-shaped chevrons and have coarser serrations. Therefore, we are confident in assigning these specimens to Otodus megalodon. Additionally, we found that relatively few Otodus megalodon teeth from ENP Neogene sediments are present in museum collections; for example, a total of 145 teeth from lower Miocene through Pliocene west coast deposits are represented in LACM, SDNHM, and UCMP collections, primarily from California. In comparison, Purdy et al. (2001:131) referred 82 specimens in addition to “several hundred isolated teeth” from the Pungo River Limestone and Yorktown Formation at the Lee Creek mine alone, and countless additional teeth exist in other collections and from other stratigraphic units from the Atlantic coastal plain. Intense collecting at ENP localities like the Sharktooth Hill Bonebed suggests that this is not simply a case of collection bias and likely reflects genuine rarity (whether biogenic or taphonomic) of Otodus megalodon teeth from Pacific coast deposits. An alternative hypothesis is a geochronologically earlier extinction of Otodus megalodon in the Pacific basin than the Atlantic.

Capistrano Formation

Stratigraphy: A thick section of late Neogene mudrock exposed in Orange County, California, are divided into the Monterey/Temblor Formation (early late Miocene) and the Capistrano Formation (latest Miocene to early Pliocene). In southern Orange County, the Capistrano Formation is between 300 and 650 m thick, and includes a basal turbidite unit composed of breccia, sandstone, and siltstone, and an upper micaceous siltstone unit (Vedder, 1972; Ingle, 1979). The Oso Member of the Capistrano is a coarse clastic tongue within the finer grained parts of the Capistrano (not formally named as member) interpreted as the distal deposits of a delta within a shallow embayment (Vedder, Yerkes & Schoellhamer, 1957; Barboza et al., 2017).

Occurrence: Specimens recovered from the Capistrano Formation (latest Miocene—early Pliocene) include LACM 59836, 58937, 115989, 129982, and SDNHM 53167 (Fig. 2). SDNHM 53167 is an incomplete upper left anterior tooth and represents the largest specimen from the Capistrano Formation (Figs. 2A and 2B). The other specimens from the Capistrano Formation represent both anterior and posterolateral teeth and range from nearly complete (LACM 129982, Figs. 2C and 2D; LACM 115989, Figs. 2G and 2H) to highly fragmented (LACM 59837, Figs. 2E and 2F; LACM 59836, Figs. 2I and 2J). SDNHM 53167 was collected from the upper siltstone unit of the Capistrano Formation (SDNHM locality 3842) from a horizon approximately 30 m below a bed which yielded diatoms of the earliest Pliocene Thalassiosira oestruppi zone (T.A. Deméré, 2012, personal communication; Deméré & Berta, 2005), dated at approximately 5.6–3.7 Ma in age (Barron & Gladenkov, 1995; Barron & Isaacs, 2001).

Figure 2 Otodus megalodon teeth from the Capistrano Formation.

SDNHM 53167 in lingual (A) and labial (B) view; LACM 129982 in lingual (C) and labial (D) view; LACM 59837 in lingual (E) and labial (F) view; LACM 115989 in lingual (G) and labial (H) view; LACM 59836 in lingual (I) and labial (J) view.

Age Conclusion: This occurrence of Otodus megalodon can be best summarized as latest Miocene to earliest Pliocene in age (latest Messinian to Zanclean equivalent, 5.6–3.7 Ma). Other specimens from the Capistrano Formation (LACM 58936, 59837, 115989, 129982) were collected from unknown horizons within the Capistrano Formation. A record of Otodus megalodon was listed by Pimiento & Clements (2014: table S1) from the Capistrano Formation and dated to 11.6–3.6 Ma, without explanation or an accompanying Paleobiology Database entry. Specimens reported from the Oso Member of the Capistrano Formation by Barboza et al. (2017) are 6.6–5.8 Ma in age (Messinian) based on the occurrence of the extinct horse Dinohippus interpolatus.

Fernando Formation

Stratigraphy: The Fernando Formation of Eldridge & Arnold (1907) is a poorly defined unit of Pliocene marine sediments in the Ventura and Los Angeles basins of southern California (Eldridge & Arnold, 1907; Woodring, Bramlette & Kew, 1946; Vedder, 1972; Squires, 2012). The Fernando Formation unconformably overlies several Miocene units, including the terrestrial Sycamore Canyon Member of the Puente Formation and the marine Capistrano and Monterey Formations (Vedder, 1972) in Orange County. The Fernando Formation was defined only on biostratigraphic age and includes numerous lithologies (Eldridge & Arnold, 1907; Squires, 2012). Because of confused relationships with other late Neogene marine rocks in southern California (e.g., Pico, Towsley, and Repetto Formations) and poor exposure, the stratigraphy and age of various outcrops assigned to the Fernando Formation remains uncertain.

Occurrence: Eldridge & Arnold (1907) listed a single occurrence of Otodus megalodon (as Carcharodon rectus, a junior synonym of Otodus megalodon) from the Shatto Estate locality; however, no photograph, specimen number, or repository information was given and thus it is not possible to unambiguously interpret this record. Eldridge & Arnold (1907) also reported the shark I. planus (as Oxyrhina plana) in addition to numerous mollusks indicating a late Pliocene to Middle Pleistocene age (C.L. Powell II, 2013, personal communication). I. planus is only represented in upper Oligocene through lower upper Miocene sediments (Chattian-Tortonian equivalent; Boessenecker, 2011b:14). The lack of reliable provenance casts doubt on the validity of this record and the reported presence of I. planus suggests a Miocene age; this record is not considered further.

Three teeth are recorded from the Fernando Formation (Fig. 3), including two specimens (LACM 148311 and 148312) from Eagle Glen in Riverside County (LACM locality 7321) and a single specimen (LACM 149739) from nearby LACM locality 7481. LACM 148311 and 148312 are fragmentary with thin and abraded enameloid, and the serrations have been eroded away. LACM 149739 is now missing, though an existing photograph shows this specimen is fragmented yet exhibits unabraded cutting edges.

Figure 3 Otodus megalodon teeth from the Fernando Formation.

LACM 148312 in lingual (A) and labial (B) view; LACM 148311 in lingual (C) and labial (D) view.

Conclusion: Owing to poor understanding of the lithostratigraphy and age of the Fernando Formation at this locality, the age of these specimens—whether reworked or not—is equivocal, and the age of the Fernando Formation is best summarized as Pliocene to Pleistocene. Accordingly, this record was excluded from the OLE.

Imperial Formation

Stratigraphy: The Imperial Formation is a thick succession of fossiliferous mudrocks deposited on the west side of the Salton Trough in Imperial County, California, and exposed best in the vicinity of the Coyote, Fish Creek, and Vallecito Mountains (Powell, 1988; Winker & Kidwell, 1996; Dorsey et al., 2011). The Imperial Formation was upgraded to group rank and subdivided into the Latrania and Deguynos members by Winker & Kidwell (1996), though we follow Powell (2008) in not recognizing this as the redefined units have not been formally described. The Fish Creek-Vallecito section of the Imperial Formation, the underlying Split Mountain Group (Miocene), and the overlying Palm Springs Group (Pliocene-Pleistocene) has been the focus of extensive stratigraphic studies investigating regional tectonics, magnetostratigraphy, biostratigraphy, and the formation of the Colorado River (Dorsey et al., 2011). The Imperial Formation has long been considered Pliocene (Hanna, 1926; Durham, 1954), and recent magnetostratigraphic work on the Fish Creek-Vallecito section indicates that the Imperial Formation spans chrons C3An.1r to C3n.1n, indicating an age of 6.43–4.187 for the entire unit (Dorsey et al., 2011).

Occurrence: A single tooth (USNM 324542) of “Carcharodon arnoldi” was reported by Hanna (1926) from the Imperial Formation at Garnet Cañon (USGS locality 3922) in the Coyote Mountains of Imperial County, California, near the type locality of the extinct walrus Valenictus imperialensis (Mitchell, 1961). This specimen is clearly a tooth of Otodus megalodon owing to its large size and clearly preserved V-shaped dental band. It is unclear whether this came from the lower (Latrania) or upper (Deguynos) member of the formation.

Conclusion: By extrapolating magnetostratigraphy from the better studied Fish-Creek Vallecito section, this occurrence of Otodus megalodon is latest Miocene to early Pliocene (6.43–4.187 Ma).

Lomita Marl

Stratigraphy: The Lomita Marl consists mostly of unconsolidated calcareous mudrocks and sandstones exposed in the western Los Angeles basin in the vicinity of Torrance and Lomita northeast of the Palos Verdes Hills, Los Angeles County, California (Grant & Gale, 1931; Woodring, Bramlette & Kew, 1946; Fig. 1). The Lomita Marl is, in part, a lateral and temporal equivalent of the Timms Point Silt and the San Pedro Sand (Woodring, Bramlette & Kew, 1946). Glauconite K/Ar dates of 3.0 Ma reported by Obradovich (1965) suggest a Pliocene age, though most modern workers consider the Lomita Marl to be middle Pleistocene in age based on aminostratigraphy and magnetostratigraphy (Ponti, 1989; Dupré et al., 1991; Lajoie et al., 1991).

Occurrence: Otodus megalodon is represented from this unit by teeth of “Carcharodon branneri” Jordan, 1922 (RMM A597-1, A597-12) and “Carcharodon leviathan” Jordan, 1922 (RMM A597-9A, A597-9B), both junior synonyms of Otodus megalodon; these specimens were collected in a quarry that exposed the unconformable contact between the Miocene Monterey Formation and Pleistocene Lomita Marl. These specimens are fragmented, abraded, with polished enameloid and phosphatic matrix adhering in cracks. Mount (1974) noted that several marine vertebrate fossils appear to be reworked from underlying Miocene rocks, supported by the preservation of these specimens (Boessenecker, Perry & Schmitt, 2014).

Conclusion: These specimens appear to have been reworked or anthropogenically mixed with middle Pleistocene sediment approximately 650–350 Ka in age (See Purported Pleistocene and Holocene records of Otodus megalodon).

Niguel Formation

Stratigraphy: The Niguel Formation is a unit of unconsolidated conglomerates, sandstones, and siltstones exposed in the San Joaquin Hills in Orange County, California, deposited along the southeastern margin of the Los Angeles Basin and unconformably overlying the Capistrano Formation and other strata (Vedder, 1972). At SDNHM locality 4080, the Niguel Formation unconformably overlies the lower-middle Miocene “Topanga” Formation (T.A. Deméré, 2013, personal communication). The base of the Niguel Formation is a conglomerate lag deposit (Vedder, 1972). The Niguel Formation is rich in fossils and mollusks indicating a Pliocene age (Vedder, 1972) possibly between 3.3 and 3.15 Ma (Powell, Stanton & Liff-Grier, 2008). Based on fossils and lithostratigraphy, Ehlig (1979) considered the Niguel Formation to be late Pliocene to Pleistocene in age, estimating it to be one to three Ma (Kern & Wicander, 1974; Powell, Stanton & Liff-Grier, 2008).

Occurrence: An abraded tooth fragment identifiable as Otodus megalodon (SDNHM 73462) was collected from the basal conglomerate, along with teeth of other sharks including Carcharhinus sp., Carcharodon carcharias, C. hastalis, Galeocerdo sp., Hemipristis sp., I. planus, and Myliobatis sp. Also recovered from this locality were tooth fragments of Desmostylus sp., earbones of a delphinid dolphin and a balaenid mysticete, and a pharyngeal tooth plate of the sheepshead fish Semicossyphus. Another Otodus megalodon specimen, LACM 59065 from Capistrano Highlands (LACM locality 65187), likely represents an upper anterior tooth (Figs. 4A and 4B) and exhibits longitudinal cracks, abraded cutting edges, and a fragmented root.

Figure 4 Otodus megalodon tooth from the Niguel Formation.

LACM 59065 in lingual (A) and labial (B) view.

Although certain marine vertebrates from SDNHM locality 4080 such as Carcharodon carcharias and Delphinidae indet. are consistent with a Pliocene age for the Niguel Formation, several other taxa are typical of older Miocene age. For example, the youngest records of desmostylians occur in the Tortonian equivalent Santa Margarita Sandstone in Santa Cruz County, and the Monterey Formation in Orange County, California (Mitchell & Repenning, 1963; Barnes, 1978, 2013; Domning, 1978). Other Miocene vertebrates from this locality include C. hastalis and I. planus; C. hastalis is replaced by C. hubbelli at approximately 8–7 Ma (Ehret et al., 2012), whereas confirmable records of I. planus are Tortonian and older (Boessenecker, 2011b:14).

Conclusion: The taphonomic condition of these Otodus megalodon specimens and presence of strictly Miocene marine vertebrates, and the occurrence of these specimens in the basal conglomerate of the Niguel Formation all indicate they were reworked from the early to middle Miocene “Topanga” Formation. Accordingly, this record was excluded from the OLE.

Purisima Formation

Stratigraphy: The Purisima Formation comprises a series of lightly consolidated sandstones, mudrocks, and diatomites of latest Miocene and Pliocene age representing shoreface to offshore sedimentation, and is exposed mostly west of the San Andreas fault in the vicinity of Santa Cruz, Halfmoon Bay, and Point Reyes in central and Northern California (Cummings, Touring & Brabb, 1962; Norris, 1986; Powell, 1998; Powell et al., 2007; Boessenecker, Perry & Schmitt, 2014). The Purisima Formation is richly fossiliferous, including fossils of sharks, bony fishes, marine birds, and marine mammals (see Boessenecker, 2011b, 2013b; Boessenecker, Perry & Schmitt, 2014, and references therein).

Occurrence: A single nearly complete upper anterior tooth of Otodus megalodon (UCMP 219502; Fig. 5) was reported by Boessenecker (2016) from the basal bonebed of the Miocene to Pliocene Purisima Formation near Santa Cruz, California (UCMP locality V99875). Only the root lobes and a small portion of the crown base are missing, and longitudinal enameloid cracks are evident lingually and labially. The basal meter of the Purisima Formation is composed of glauconitic sandstone and a matrix-supported conglomerate with abundant vertebrate skeletal elements mantling an erosional surface with ∼1 m of relief, unconformably overlying the upper Miocene Santa Cruz Mudstone (Clark, 1981; Boessenecker, Perry & Schmitt, 2014). Glauconite from the base of the Purisima Formation has yielded a K/Ar date of 6.9 ± 0.5 Ma (Clark, 1966; Powell et al., 2007). A tuff bed approximately 30 m above the base of the Purisima Formation has been tephrochronologically correlated with 5.0 ± 0.3 Ma tephra in the Pancho Rico Formation (Powell et al., 2007).

Figure 5 Otodus megalodon tooth from the Purisima Formation.

UCMP 219502 in lingual (A) and labial (B) view.

Conclusion: This locality (UCMP locality V99875) can be summarized as 6.9–5.3 Ma in age, or latest Miocene (Messinian equivalent).

San Diego Formation

Stratigraphy: The San Diego Formation comprises approximately 85–90 m of unconsolidated Pliocene and Pleistocene sandstones, mudrocks, and conglomerates of terrestrial and marine origin deposited via extensional tectonics within a graben in the vicinity of San Diego, California between Pacific Beach and northern Baja California (Deméré, 1982, 1983; Wagner et al., 2001; Vendrasco et al., 2012). The San Diego Formation is informally divided into two members: a “lower” sandstone member that is entirely marine in origin, and an “upper” sandstone and conglomeratic member that is marine and terrestrial (Deméré, 1982, 1983). Although earlier studies concluded that the San Diego Formation was approximately 3–1.5 Ma in age (late Pliocene to early Pleistocene: Deméré, 1983), more recent estimates based on paleomagnetism and correlation with patterns of eustatic sea level change suggest an early Pliocene age (Zanclean equivalent) for parts of the “lower” member of the San Diego Formation (Wagner et al., 2001). Furthermore, Vendrasco et al. (2012) reported the San Diego Formation to ranges in age from 4.2 to 1.8 Ma.

Occurrence: A single upper right anterior or anterolateral tooth missing part of the root and crown (SDNHM 29742; Figs. 6A and 6B) was reported from the basal San Diego Formation near La Joya in Baja California (SDNHM locality 3253; Ashby & Minch, 1984). The tooth is almost equilateral, with a slight curvature to the right. A V-shaped chevron, fine serrations, and three small nutrient foramina are present on the lingual surface of the root. Three additional specimens (Figs. 6C–6H) are recorded from LACM collections from San Diego County: LACM 156334 (LACM locality 1095), a broken tooth with thinned and longitudinally cracked enameloid, abraded surfaces and broken edges; LACM 10152 (LACM locality 4875), a broken but unabraded tooth with longitudinally cracked enameloid; LACM 103448 (LACM locality 1080), a fragment of enameloid shell missing the orthodentine core. These other specimens are less complete than SDNHM 29742 are not stratigraphically located within the San Diego Formation.

Figure 6 Otodus megalodon teeth from the San Diego Formation.

SDNHM 29742 in lingual (A) and labial (B) view; LACM 156334 in lingual (C) and labial (D) view; LACM 10152 in lingual (E) and labial (F) view; LACM 103448 in lingual (G) and labial (H) view.

Conclusion: The only specimen with precise stratigraphic data (SDNHM 29742) was collected from the basal unconformity of the San Diego Formation. This occurrence can be summarized as approximately 4.2 Ma in age (early Pliocene); a minimum age of 3.6 Ma is provided by magnetostratigraphy of strata higher up in the San Diego Formation (Wagner et al., 2001).

San Mateo Formation

Stratigraphy: The San Mateo Formation is a thin package of unconsolidated sandstones and conglomerates, which crop out in the vicinity of Oceanside in San Diego County, California. It is considered a temporal equivalent of the Oso Member of the Capistrano Formation (Barnes et al., 1981; Domning & Deméré, 1984), and Vedder (1972) refers to it as a coarse clastic tongue within the Capistrano Formation. It consists of a lower unit composed of massive, fine-grained sandstones with occasional muddy lenses, sparse pebbles and cobbles, and an upper unit of complexly bedded sandstones and conglomerates; a sharp erosional surface at the base of the upper unit divides the formation (Barnes et al., 1981; Domning & Deméré, 1984). Fossil assemblages from the lower and upper units have been termed the San Luis Rey River and Lawrence Canyon local faunas, respectively (Barnes et al., 1981). Domning & Deméré (1984) interpreted the lower unit to represent middle or inner shelf deposition, and the upper unit to represent the distal margin of a submarine fluvial delta system. A diverse marine vertebrate assemblage including sharks, bony fish, marine birds, and marine mammals is now known from the San Mateo Formation at Oceanside (Barnes et al., 1981; Domning & Deméré, 1984; Long, 1994). Due to the lack of macroinvertebrates or microfossils, age estimates for the San Mateo Formation have been established based on vertebrate biochronology, including terrestrial mammals and mancalline auks (Domning & Deméré, 1984). Barnes et al. (1981) considered both the lower and upper units to be correlative with the Hemphillian NALMA. However, Domning & Deméré (1984) reported that the presence of Aepycamelus indicated the lower unit is slightly older, perhaps late Clarendonian to early Hemphillian in age (approximately 10–7 Ma; Tedford et al., 2004), and correlated the upper unit with the late Hemphillian NALMA (7 Ma to 4.9–4.6 Ma; Tedford et al., 2004). Based on the presence of mancalline auks found in other rocks of early Pliocene age (and the lack of late Pliocene mancalline taxa as from the San Diego Formation), Domning & Deméré (1984) indicated an early Pliocene age for the upper unit of the San Mateo Formation.

Occurrence: The San Mateo Formation has yielded a number of partial Otodus megalodon teeth including: SDNHM 23056, 23959 (several teeth in a lot), 24448, 77430, 77343, and LACM 131149 (Fig. 7). One specimen catalogued in the lot SDNHM 23959 (Figs. 7I and 7J) and another tooth (SDNHM 24448, Figs. 7C and 7D) represent the most complete teeth recovered from the San Mateo Formation. SDNHM 23959 represents an upper right anterolateral tooth consistent with Otodus megalodon despite missing the apex, having worn and chipped mesial and distal cutting edges, and broken root lobes. SDNHM 24448 represents an upper left posterolateral tooth (Figs. 7C and 7D). The specimen is missing a portion of the right root lobe and is missing some enameloid on the lingual surface of the crown.

Figure 7 Otodus megalodon teeth from the San Mateo Formation.

LACM 131149 in lingual (A) and labial (B) view; SDNHM 24448 in lingual (C) and labial (D) view; SDNHM 23959 in lingual (E) and labial (F) view; SDNHM 77343 in lingual (G) and labial (H) view; SDNHM 23959 in lingual (I) and labial (J) view; SDNHM 23959 in lingual (K) and labial (L) view; SDNHM 23959 in lingual (M) and labial (N) view.

Conclusion: Teeth of Otodus megalodon occur in both the lower and upper units of the San Mateo Formation (Domning & Deméré, 1984; Barnes & Raschke, 1991), and occurrences from the upper unit are here summarized as earliest Pliocene in age (5.33–4.6 Ma).

Santa Cruz Mudstone

Stratigraphy: At the type section west of Santa Cruz (Santa Cruz County) of the Santa Cruz Mudstone is a monotonous succession of jointed, indurated, and siliceous mudrocks (siltstone and porcelanite), which conformably overlies the Santa Margarita Sandstone and is in turn unconformably overlain by the Purisima Formation. In the vicinity of Point Reyes, thick, massively bedded, indurated, and fractured siliceous mudrocks were originally considered by Galloway (1977) to represent both the Monterey and Drakes Bay formations, but were remapped by Clark et al. (1984) as the somewhat younger Santa Cruz Mudstone. Near Bolinas, foraminifera representative of the Delmontian California benthic foraminiferal Stage (∼7–5 Ma; Barron & Isaacs, 2001) has been recorded, in addition to a diatom flora typical of Diatom Zone X (Clark et al., 1984), which was later refined to subzone A of the Nitzschia reinholdii zone by Barron (in Zeigler, Chan & Barnes, 1997), equivalent to 7.6–6.5 Ma (Barron & Isaacs, 2001). Fossil bivalves from the Santa Cruz Mudstone at Bolinas indicate deposition at about >500 m depth (Zeigler, Chan & Barnes, 1997). Fossil vertebrates from the Santa Cruz Mudstone include the baleen whale Parabalaenoptera baulinensis (Zeigler, Chan & Barnes, 1997), the sea cow Dusisiren dewana (initially reported as Dusisiren species D by Domning, 1978), a herpetocetine baleen whale (Boessenecker, 2011a:8), and a number of unpublished marine mammals (R.W. Boessenecker, 2015, personal observation) including a phocoenid porpoise (cf. Piscolithax), an albireonid dolphin, fragmentary odobenid and otariid bones, and earbones of indeterminate balaenopterid mysticetes.

Occurrence: A single tooth of Otodus megalodon was reported from “Bolinas Bay” by Jordan (1907: figure 15) as the holotype specimen of “Carcharodon branneri.” Unfortunately, searches for additional locality information at CAS were unsuccessful. Specimens reported by D.S. Jordan in various publications were originally curated at Stanford University, the collections of which were later transferred to CAS; it is possible that some of these specimens were never transferred to CAS (S. Mansfield, 2013, personal communication; D. Long, 2013, personal observation). Ransom (1964) published township and range coordinates for this locality, suggesting that the type was collected near the west shore of the Bolinas Lagoon in the vicinity of the Bolinas County Park. However, this area is covered by Quaternary alluvium with nearby exposures of sparsely fossiliferous Pliocene to Pleistocene Merced Formation. It is more likely that the original locality information is correct, and that the type specimen was collected from exposures (or as float) of the Santa Cruz Mudstone along the northwestern shore of Bolinas Bay (Jordan & Hannibal, 1923; also see Jordan, 1907) or possibly from as far west as Duxbury Reef (where the majority of twentieth and twenty-first century vertebrate collections from this unit have been made). This specimen was erroneously assigned to the Purisima Formation by Pimiento & Clements (2014: table S2) and assigned an age of 5.3–2.6 Ma without explanation; the Purisima Formation does not crop out anywhere within 25 km of Bolinas (Clark et al., 1984). Bones of fossil marine mammals are often collected as float from these beaches.

Conclusion: If this specimen was collected from the Santa Cruz Mudstone near Bolinas, then it likely represents an older 7.6–6.5 Ma record. Because this record falls within the Messinian, it was included within the OLE.

Tirabuzón Formation

Stratigraphy: The Tirabuzón Formation consists of unconsolidated fossiliferous sandstone exposures in the vicinity of Santa Rosalia along the eastern side of the northern Baja California Peninsula (Applegate, 1978; Applegate & Espinosa-Arrubarrena, 1981; Wilson, 1985). Formerly mapped as the Gloria Formation, it was renamed the Tirabuzón Formation by Carreno (1982) after abundant spiral burrows of the ichnogenus Gyrolithes which leant the locality the name “Corkscrew Hill.” Paleodepth estimates for this unit range from 200 to 500 m (outer shelf to slope) based on foraminifera (Carreno, 1982) to 55–90 m (middle shelf) based on ichnology (Wilson, 1985). The Tirabuzón Formation unconformably overlies the upper Miocene Boleo Formation, and is in turn unconformably overlain by the lower to upper Pliocene Infierno Formation (Holt, Holt & Stock, 2000). Holt, Holt & Stock (2000) reported an 40Ar/39Ar date of 6.76 ± 0.9 Ma from an andesitic interbed within the Boleo Formation, constraining a lower limit for the age of the Tirabuzón Formation. The age of the Tirabuzón Formation was considered Pliocene by Applegate (1978) and Applegate & Espinosa-Arrubarrena (1981), and approximately 4–3 Ma (Zanclean equivalent) by Barnes (1998). Mollusks reported from the overlying Infierno Formation indicate a maximum age of early Pliocene for that unit (Johnson & Ledesma-Vasquez, 2001), therefore constraining a minimum age of early Pliocene for the Tirabuzón Formation. Shark and marine mammal fossils have previously been reported from the Tirabuzón Formation near Santa Rosalia, including 34 shark taxa (including Otodus megalodon), an indeterminate otariid, two balaenopterid mysticetes, a small pontoporiid dolphin (aff. Pontoporia), an indeterminate phocoenid, two delphinids (Delphinus or Stenella sp., and aff. Lagenorhynchus sp.), two kogiids (aff. Kogia sp. and cf. Scaphokogia sp.), and an indeterminate physeterid (Applegate, 1978; Applegate & Espinosa-Arrubarrena, 1981; Barnes, 1998).

Occurrence: Small teeth of Otodus megalodon are relatively abundant in the Tirabuzón Formation (Fig. 8), and include 14 partial teeth: LACM 29064–29067, and 29069–29078. Most of these teeth, except for smaller fragments, exhibit the characteristic V-shaped chevron and most still retain their fine serrations. The most complete specimens are two left posterolateral upper teeth, LACM 29065 (Figs. 8I and 8J), missing portions of the root lobes, and LACM 29076 (Figs. 8G and 8H), missing the apex of the crown and parts of the root lobes.

Figure 8 Otodus megalodon teeth from the Tirabuzón Formation.

LACM 29067 in lingual (A) and labial (B) view; LACM 29064 in lingual (C) and labial (D) view; LACM 29077 in lingual (E) and labial (F) view; LACM 29076 in lingual (G) and labial (H) view; LACM 29065 in lingual (I) and labial (J) view; LACM 29074 in lingual (K) and labial (L) view; LACM 29069 in lingual (M) and labial (N) view; LACM 29073 in lingual (O) and labial (P) view; LACM 29075 in lingual (Q) and labial (R) view; LACM 29072 in lingual (S) and labial (T) view.

Conclusion: This occurrence of Otodus megalodon in the Tirabuzón Formation is estimated to be late Miocene to early Pliocene (Messinian-Zanclean equivalent; 6.76–3.6 Ma).

Revisions to the Pimiento & Clements (2014) dataset

Prior to conducting the OLE we thoroughly vetted every fossil occurrence in the dataset published by Pimiento & Clements (2014: table S1; Table 2). We encountered a number of issues requiring adjustments. Much of their dataset (88% of occurrences) consists of dates binned to stages (e.g., assigned the boundary dates of a particular stage or epoch). This is standard practice for paleobiological analyses (Pimiento et al., 2016) like richness counts, but it artificially expanded the age range (e.g., older maxima and younger minima) for occurrences where finer geochronological age control is available in the literature. This artificially inflated the number of Piacenzian-stage occurrences of Otodus megalodon which actually have older (e.g., Zanclean) minimum dates (n = 13 occurrences from Pimiento & Clements, 2014; Table 2). In other cases, we updated out-of-date geochronologic data; out of occurrences we did not exclude, we were able to update 25 out of 32 occurrences (78% of the dataset) originally reported by Pimiento & Clements (2014) based upon stratigraphic and geochronologic studies not cited by these authors (Table 2).

We excluded 10 out of the original 42 occurrences which satisfied one or more rejection criteria (see Materials and Methods, Appendix 2). Several occurrences did not possess subepochal age control (criterion 1), and we excluded any occurrence data for Otodus megalodon teeth where the finest age control possible was provided by a statement in the literature like “this locality is mapped as Pliocene” or “middle Miocene to Pliocene” without finer age control (Luanda Formation, Angola; Canímar Formation, Cuba; Salada Formation, Mexico). Several occurrences reported by Pimiento & Clements (2014) consist of unpublished records of Otodus megalodon teeth from the Bone Valley Formation in various mines in Florida, dated to early Pliocene; however, their stratigraphic justification refers to unpublished data in the FLMNH online data base (criterion 8). An early Pliocene age is entirely consistent with other exposures of the upper Bone Valley Formation (Morgan, 1994). However, we excluded these occurrences because the stratigraphic interpretation cannot be evaluated based upon the peer-reviewed literature alone. Other occurrences simply lacked strong provenance (criterion 4); for example, one specimen of Otodus megalodon reported by Keyes (1972) from New Zealand had locality data on a label stating “probably from the upper Miocene beds Older Wanganui Series of NZ Geological Survey from between Wanganui and N. Plymouth”; 180 km of coastline separate these two cities. We excluded this record for its lack of provenance. One purported Pliocene tooth of Otodus megalodon was reported from the Highlands Limestone of Barbuda (Flemming & McFarlane, 1998); we excluded this record because geological studies indicate this unit is actually middle Miocene in age (criterion 6) Brasier & Mather, 1975, and references therein). In one case, teeth reported from the Bahia Inglesa Formation of Chile (Long, 1993) lacked intraformational stratigraphic control (criteria 1, 2, 4), and assignment to the Miocene or Pliocene section of the formation was not possible. The age control for two other localities (Luanda Formation, Angola; “Main Vertebrate spot,” Libya) was based on biochronology of the shark assemblage, with a minimum age of early Pliocene being based on the presence of Otodus megalodon itself (Antunes, 1978; Pawellek et al., 2012). We excluded these records because inclusion of these records within the OLE would constitute circular reasoning (criterion 7). Other occurrences were excluded because they appear to be misidentified teeth of Carcharodon carcharias (Cameron Inlet Formation, Australia; Kemp, 1991: plate 30C; criterion 5) or the specimens in question still reside in a private collection and therefore cannot be evaluated by scientists (Tangahoe Formation, New Zealand; McKee, 1994; criterion 3).

We were also able to add several occurrences to the dataset (Table 2; Appendix 1), including some published after the publication of Pimiento & Clements (2014) and some that they were unaware of (e.g., Carrillo Puerto Formation, Mexico; Tokomaru Formation, New Zealand); at least one occurrence (Daito Limestone, Kita-daito-jima, Japan; Table 2) originally excluded by Pimiento & Clements (2014) was found to have stronger age control (Takayanagi et al., 2010) than previously acknowledged (Pimiento & Clements, 2014: text S1), and was included within the OLE.

Results of OLE analysis

The total range of estimates of extinction age for Otodus megalodon in the OLE analysis vary from 4.1 to 3.2 Ma, with a modal extinction date of 3.6 Ma (Fig. 9). Approximately 50% of the extinction estimates indicate extinction between 4.1 and 3.6 Ma. Fewer than 2% of the extinction estimates occur before 3.91 Ma, while fewer than 2% occur after 3.26 Ma. None of the extinction dates estimated occurred post Zanclean, with no evidence of survival of Otodus megalodon into this period or to the present.

Figure 9 Inferred dates of extinction for Otodus megalodon using the Optimal Linear Estimation (OLE) model.

Data binned by 10,000 year increments. The histogram (blue) represents the % frequency of a given date that was estimated for the extinction out of 10,000 simulations. The curve (orange) represents increasing cumulative probability that Otodus megalodon was extinct at the given date.

Discussion

Purported Pleistocene and Holocene records of Otodus megalodon

The record of Otodus megalodon from the Lomita Marl (Jordan, 1922) is substantially younger than many other records from California. However, as noted by Mount (1974), numerous sharks and other marine vertebrates from the Lomita Quarry locality are only found elsewhere in middle and late Miocene localities, such as Allodesmus (Jordan & Hannibal, 1923: plate 9J) and Carcharodon hastalis (Jordan & Hannibal, 1923: plate 9E and 9F). Furthermore, shark teeth including Otodus megalodon teeth were collected by quarry manager S. M. Purple (Bailey, 1922; Mount, 1974), without accompanying stratigraphic information, and it is unclear where in the Lomita Quarry these specimens were collected. Hanna (in Jordan & Hannibal, 1923) notes that the base of the Lomita Marl within the Lomita Quarry was a glauconitic sandstone with abundant abraded whale bones, and that in addition to Miocene marine mammals and sharks, Pleistocene terrestrial mammals and a single Pleistocene pinniped were present in the quarry. This curious mixture of taxa suggests stratigraphic reworking of older fossil material; indeed, the holotype specimen of the gastropod Mediargo mediocris was considered by Wilson & Bing (1970:7) to be reworked from Pliocene sediments into the Lomita Marl. Woodring, Bramlette & Kew (1946) report that the Lomita Marl includes “beds of gravel consisting chiefly or entirely of limestone pebbles and cobbles derived from the “Monterey” Shale. Locally huge boulders of soft Miocene mudstone and Pliocene siltstone are embedded in calcareous strata.” These specimens of Otodus megalodon (RMM A597-1, A597-9A, and A597-9B) are fragmented, and strongly abraded with polished enameloid, indicating reworking. Only RMM A597-12 showed little evidence of abrasion, although taphonomic experiments on fossil teeth by Argast et al. (1987) noted that abrasion is not a guaranteed outcome of transport or reworking. Lastly, anthropogenic mixing of multiple strata during mining operations is also a possibility for seemingly older taxa in younger beds. Dynamite was used for mining in the quarry, which apparently “[brought] down bones of whales, sea lions, land animals, chipped flints, pieces of charcoal, sea shells, shark’s teeth, arrowheads, all mixed together” (Bailey, 1922). The report of Otodus megalodon from the Pleistocene Lomita Marl could be due to reworking from the “Monterey” Formation, anthropogenic mixing from mining operations, collection from underlying strata, poor record keeping, or any combination of the above. In this context, teeth of Otodus megalodon from the Lomita Marl are considered to be allochthonous (either by sedimentologic or anthropogenic reworking) and thus not relevant to the consideration of the timing of the extinction of the species.

Three teeth of Otodus megalodon (LACM 10141, 11194, and 159028) are questionably recorded from the upper Pleistocene Palos Verdes Sand and unnamed strata at Newport Bay Mesa (Fig. 10). LACM 11194 is now missing, but was found by an unknown collector prior to 1915 from the North Pacific Avenue and Bonita Avenue intersection in northern San Pedro, California. The locality is now built over, but was mapped as Palos Verdes Sand by Woodring, Bramlette & Kew (1946). LACM 10141, is a fragmentary tip of a tooth with longitudinally cracked enameloid and abraded serrations (Figs. 10C and 10D), and was collected from unnamed strata along the Newport Bay Mesa formerly considered to belong to the Palos Verdes Sand (collector and collection date unknown); it is alternatively possible that this specimen was collected from an exposure (now covered) of the Pliocene Fernando Formation (see Mount, 1969). LACM 159028 (Figs. 10A and 10B) possesses the following dubious locality information: “Rosecranz Ave. Long Beach, Orange Co.?” We note that Rosecrans Avenue is far from the Palos Verdes Hills and from Long Beach, and that both Rosecrans Avenue and Long Beach are located within Los Angeles County. It is also possible that this specimen is reworked from the underlying Puente Formation (L.G. Barnes, 2015, personal communication). Alternatively, some Pliocene rocks are known in the Coyote Hills near Rosecrans Avenue (Powell & Stevens, 2000), and the specimen may have been collected there. It is not possible to unambiguously recognize any of these specimens as genuine Pleistocene records of Otodus megalodon given the lack of provenance. We also note the similarity in preservation (chiefly color) between LACM 159028 and teeth of Otodus megalodon from some localities at Sharktooth Hill (middle Miocene “Temblor” Formation, Kern County). Kanakoff (1956) only listed C. carcharias from this unit. Furthermore, a comprehensive study of the ichthyofauna by Fitch (1970) only recorded C. carcharias. We hypothesize that LACM 11194 was a misidentified or mistranscribed specimen of C. carcharias and that the other two specimens originated from a separate locality. Therefore, we conclude that no reliable records of Otodus megalodon exist for Pleistocene deposits in the Los Angeles Basin.

Figure 10 Otodus megalodon teeth of purported Pleistocene age.

LACM 159028 in lingual (A) and labial (B) view, supposedly from Palos Verdes Sand; LACM 10141 in lingual (C) and labial (D) view, supposedly from unnamed strata at Newport Bay Mesa.

Several studies have reported teeth of Otodus megalodon dredged from the seafloor and considered to be Pleistocene or even Holocene in age (Tschernezky, 1959; Seret, 1987; Roux & Geistdoerfer, 1988). Dredged specimens from the south Pacific were reported by Tschernezky (1959) and Seret (1987), whereas Roux & Geistdoerfer (1988) reported numerous specimens from the Indian Ocean seafloor off the coast of Madagascar. Tschernezky (1959) and Roux & Geistdoerfer (1988) both attempted to determine the age of the teeth by measuring the thickness of adhering manganese dioxide nodules and applying published rates of MnO2 nodule growth. Tschernezky (1959) reported a range of 24,406–11,333 years for the MnO2 nodule formation for these teeth, and Roux & Geistdoerfer (1988) reported specimens with nodules with the equivalent of 60–15 Ka of MnO2 growth. However, both studies assumed a constant rate of nodule growth and interpreted these dates as indicating a latest Pleistocene extinction of Otodus megalodon (Tschernezky, 1959; Roux & Geistdoerfer, 1988). Tschernezky (1959) argued that even if Otodus megalodon went extinct during the Middle Pleistocene ca. 500 Ka, his dredged Otodus megalodon teeth should have had MnO2 coatings approximately 75 mm thick. Because conditions favoring the formation and growth of MnO2 nodules are not necessarily constant over geologic time (Purdy et al., 2001), these dates can only indicate when these teeth were exposed to seawater and do not reflect geochronologic age. It is probable that these specimens were concentrated on the seafloor through submarine erosion, winnowing, depositional hiatus, or a combination thereof. Collections of numerous resistant vertebrate hardparts from these dredgings (e.g., shark teeth and cetacean ear bones) support this suggestion. A more parsimonious scenario is that these specimens are Pliocene (or Miocene) in age and were deposited in areas of slow sedimentation with intermittent erosion, concentrating nodules and resistant marine vertebrate skeletal elements (typically teeth and cetacean skull fragments) on the seafloor. Intermittent periods of favorable chemistry fostered the formation and growth of MnO2 nodules and coatings, and it is possible that these specimens have experienced numerous burial-exhumation cycles (Boessenecker, Perry & Schmitt, 2014). Lastly, because no extrinsic absolute or biostratigraphic age data exist for these specimens, the maximum age of these specimens is ultimately unknown and cannot be considered to represent post-Pliocene occurrences (Applegate & Espinosa-Arrubarrena, 1996; Purdy et al., 2001).

Timing of the extinction of Otodus megalodon in the eastern North Pacific

Although numerically less abundant than in deposits of the Atlantic Coastal Plain, fossil teeth of Otodus megalodon have been reported from numerous middle Miocene localities in California and Baja California (Jordan & Hannibal, 1923; Mitchell, 1966; Deméré et al., 1984). Late Miocene occurrences of Otodus megalodon include the Almejas (Barnes, 1992), Monterey (Barnes, 1978), “lower” San Mateo (Domning & Deméré, 1984), Capistrano (Barboza et al., 2017; this study), and Purisima formations (Boessenecker, 2016; this study), Santa Cruz Mudstone (Jordan & Hannibal, 1923; this study), and Santa Margarita Sandstone (Barnes, 1978; Domning, 1978). Pliocene occurrences in California (reviewed above) are restricted to the Capistrano, Fernando, “upper” San Mateo, basal San Diego, and Tirabuzón formations (Fig. 11). In the context of dubious provenance or clear evidence of reworking for specimens younger than these, we do not consider any post-early Pliocene records of Otodus megalodon to be reliable and putative Quaternary specimens are particularly dubious. Several specimens of Otodus megalodon are now recorded from the basal San Diego Formation, which is as old as 4.2 Ma (Wagner et al., 2001; Vendrasco et al., 2012), and we interpret these records as earliest Pliocene (Zanclean equivalent; Fig. 11). The lack of Otodus megalodon specimens and abundant Carcharodon carcharias teeth in younger sections of the San Diego Formation is paralleled in the Purisima Formation at Santa Cruz. Although Carcharodon carcharias teeth are common within well-sampled bonebeds, no teeth of Otodus megalodon have been discovered from the Pliocene section of either unit. However, teeth of Otodus megalodon are rare within established Miocene marine vertebrate collections relative to C. hastalis or C. carcharias (e.g., within the Sharktooth Hill Bonebed, approximately 80+ specimens of C. hastalis are recorded vs. nine teeth of Otodus megalodon in UCMP collections; accessed October 23, 2018). In summary, specimens discussed herein are entirely latest Miocene or earliest Pliocene in age (Messinian-Zanclean equivalent; Fig. 11). Qualitative assessment of the reliable occurrences of Otodus megalodon in California and Baja California suggests extinction of this taxon during the early Pliocene, perhaps during the Zanclean stage or near the Zanclean-Piacenzian boundary (ca. 4–3 Ma; Fig. 11).

Figure 11 Geochronologic age range of Otodus megalodon-bearing strata and occurrences in the eastern North Pacific.

Age control of latest Miocene and Pliocene Otodus megalodon-bearing stratigraphic units represented by thick vertical gray bars. Stratigraphic range of autochthonous and parautochthonous Otodus megalodon occurrences (allochthonous records excluded) depicted as thin vertical black bars. Abbreviations: NALMA, North American Land Mammal Age.

A worldwide view of Otodus megalodon extinction

The fossil record of Otodus megalodon in other regions lends support to an early Pliocene (Zanclean) extinction (Figs. 9 and 11). Previously described records of Pliocene age possibly relevant to temporally constraining the extinction of Otodus megalodon include occurrences from the eastern USA, Japan, Australia, New Zealand, western Europe (Belgium, Spain, UK, Denmark), southern Europe (Italy), Africa (Libya), and South America (Chile, Ecuador, Peru, Venezuela).

In deposits around the North Sea, Otodus megalodon has been reported from the Miocene, Pliocene, and Pleistocene (Bendix-Almgreen, 1983; Donovan, 1988). A tooth from the upper Miocene Gram Formation of Denmark was interpreted by Bendix-Almgreen (1983:23–24) as representing the youngest record of Otodus megalodon from the eastern North Atlantic. A tooth of Otodus megalodon from the Pliocene to Pleistocene Red Crag Formation of eastern England was mentioned by Donovan (1988), although the majority of marine vertebrate remains—marine mammals in particular—show evidence of reworking including abrasion, polish, and phosphatization and furthermore typically consist of dense elements with relatively high preservation potential (e.g., cetacean tympanoperiotics, teeth and tusks, and osteosclerotic beaked whale rostra; Owen, 1844, 1870; Lydekker, 1887). This evidence suggests that marine vertebrate material has been reworked from preexisting strata predating the Red Crag Formation; indeed, the Red Crag unconformably overlies the Eocene London Clay and the lower Pliocene Coralline Crag Formation (Zalasiewicz et al., 1988), and marine vertebrate remains may date to the Eocene-Pliocene depositional hiatus (or erosional lacuna) between the London Clay and overlying Red Crag Formation, or may have been reworked from the Coralline Crag Formation. A single record from the Piacenzian of France is cited by Cappetta (2012) from Gervais (1852), but no geographic or stratigraphic information is given by Gervais (1852:173) and this record cannot be evaluated.

In a review of the stratigraphic range of Pliocene to Pleistocene elasmobranchs from Italy, Marsili (2008) indicated that Otodus megalodon disappeared from the record during the Zanclean (∼4 Ma) and that no Piacenzian records existed, contra Pimiento & Clements (2014: table S1). In their discussion of the shark fauna of Malta, Ward & Bonavia (2001) considered Otodus megalodon to have become extinct in the early Pliocene (but without further comment). Other early Pliocene (Zanclean equivalent) records of Otodus megalodon from western Europe and the Mediterranean region include the Huelva Formation of Spain (Garcia et al., 2009) and unnamed strata in the Sabratah Basin of northwestern Libya (Pawellek et al., 2012). Elsewhere in Africa, Otodus megalodon is recorded from the early Pliocene of Angola (Antunes, 1978).

In a summary of Mesozoic and Cenozoic ichthyofaunas from Japan, Yabumoto & Uyeno (1995) reported that Otodus megalodon is widely known from Miocene strata and occurs in the lower Pliocene, but not from younger upper Pliocene and Pleistocene rocks. Subsequently, a review by Yabe, Goto & Kaneko (2004) reported widespread occurrences of Otodus megalodon in the earliest Pliocene (Zanclean) and a few late early Pliocene records (Piacenzian), and considered Otodus megalodon to have gone extinct in the late early Pliocene or late Pliocene. Three post-Zanclean occurrences were listed by Yabe, Goto & Kaneko (2004): one is uncertainly Piacenzian, another Zanclean or Piacenzian, and one strictly Piacenzian in age. However, these specimens were not figured by Yabe, Goto & Kaneko (2004) and it is unclear whether or not they are reworked.

An early Pliocene (Zanclean or Piacenzian) extinction of Otodus megalodon seems to be reflected in the fossil record of Australia and New Zealand. Late Miocene occurrences of Otodus megalodon are common from both landmasses (Keyes, 1972; Kemp, 1991; Fitzgerald, 2004). Several early Pliocene records of Otodus megalodon have been reported from Australia (Kemp, 1991; Fitzgerald, 2004), including a single specimen from the lower Pliocene Cameron Inlet Formation (Zanclean-Piacenzian correlative; Kemp, 1991; Fitzgerald, 2004). However, judging from Kemp’s (1991: plate 30C) illustration, this specimen is a misidentified Carcharodon carcharias tooth owing to its small size, lack of a preserved chevron, and relatively large serrations. Keyes (1972) reported several specimens ranging in age from early Pliocene to Pleistocene age, but many of them have tenuous provenance. For example, one such specimen (included in the analysis by Pimiento & Clements, 2014) can only be pinpointed to a 180 km section of coastline. Only a single published Pliocene tooth of Otodus megalodon from New Zealand has reliable provenance, a specimen collected from Patutahi Quarry on the North Island. According to Keyes (1972), strata at the quarry correspond to the local New Zealand Opoitian Stage (5.33–3.6 Ma); accordingly, this tooth represents the youngest reliable record of Otodus megalodon from New Zealand.

In South America, Otodus megalodon is known continuously from at least the middle Miocene to the lowermost Pliocene in the Pisco Basin of Peru (Muizon & De Vries, 1985; Ehret et al., 2012). However, owing to the absence of well-sampled younger marine vertebrate assemblages, it is unclear if this simply reflects an artifact of preservation. Otodus megalodon has also been reported from the latest Miocene-early Pliocene of Ecuador (Longbottom, 1979). Although Otodus megalodon has been reported from the well-sampled uppermost Miocene to lower Pliocene Bahia Inglesa Formation of Chile (Long, 1993), the exact age of this occurrence is imprecisely known (Walsh & Hume, 2001; Walsh & Naish, 2002). On the Caribbean coast of South America, Otodus megalodon is continuously known from middle Miocene through lower Pliocene deposits, with the youngest specimens occurring in the lowermost Pliocene (Zanclean-correlative; Aguilera, Garcia & Cozzuol, 2004).

Paralleling the record in Venezuela, abundant Miocene records of Otodus megalodon exist in the western North Atlantic and Caribbean, with the youngest specimens consistently being earliest Pliocene in age (Flemming & McFarlane, 1998; Purdy et al., 2001; Ward, 2008). In deposits of the Atlantic coastal plain of the United States, teeth of Otodus megalodon are abundant within the lower Pliocene Sunken Meadow Member of the Yorktown Formation (Purdy et al., 2001; Ward, 2008), but absent from the upper Pliocene Rushmere and Moore House members of the Yorktown Formation (Ward, 2008). The extinction of Otodus megalodon was interpreted by Ward (2008) to have occurred during the time recorded by the unconformity and depositional hiatus of uncertain duration between the Sunken Meadow and Rushmere members. A number of possible Pleistocene occurrences of Otodus megalodon from Florida are present in FLMNH collections, but originate from temporally mixed fossil assemblages and quarry spoil piles (D.J. Ehret, 2015, personal observation).

We interpret the absence of Otodus megalodon in the Rushmere and Moore House members of the Yorktown Formation, upper San Diego Formation, and “upper” parts of the Purisima Formation to be biochronologically real and reflect the genuine absence of this taxon. Given the intense collecting of these localities by amateur and professional paleontologists alike, collection bias is not likely a factor in determining the stratigraphic occurrence of Otodus megalodon.

Results of the OLE analysis based upon our revised version of the dataset (Table 2; Appendix 1) incorporating current stratigraphic and geochronologic data indicate that Otodus megalodon was most likely extinct by 3.6 Ma and perhaps even as early as 4.1 Ma (maximum) and certainly no later than 3.2 Ma (minimum; Figs. 9 and 11), strongly indicating an extinction during the Zanclean stage or close to the Zanclean-Piacenzian boundary (3.6 Ma). This global estimate compares well with the qualitative approach for Otodus megalodon occurrence data in the ENP, and consistent identification of minimum ages near the Zanclean-Piacenzian boundaries is consistent with a globally synchronous extinction at or around 3.6 Ma. We note that the separation between the maximum and minimum inferred extinction dates (∼970,000 years; 4.1–3.2 Ma) from the OLE are substantially narrower than those results (3.66 million years; 3.5 Ma–160 ky in the future) reported by Pimiento & Clements (2014:2, Fig. 1). Pimiento and Clements, in their results, focus on range of dates that result in a cumulative 50% probability of extinction, which give a range of extinction dates from 3.5 to 2.6 Ma. If we follow this reasoning, and focus on the cumulative 50% probability of extinction found in our study, we still get a narrower range of possible extinction times, between 4.1 and 3.6 Ma, or a range of 500,000 years, vs. the 900,000 of Pimiento & Clements (2014).

Perhaps more critical than the narrower range of modeled extinction dates is the earlier shift in modal extinction date by nearly one million years relative to Pimiento & Clements (2014). Geologically speaking, one million years seems trivial, but the difference between the two reconstructed extinction dates is critical given that this extinction occurred quite recently; an earlier date now poses problems for the supposed synchronicity of Plio-Pleistocene marine mammal extinctions and/or faunal turnover (Pimiento et al., 2017; but see Boessenecker, 2013a). Recently, Wang & Marshall (2016:4) noted that the “poorly resolved ages of many of the fossil occurrences” of the Pimiento & Clements (2014) dataset led a wide confidence interval in their OLE. Indeed, the results of macroevolutionary studies of extinction timing are sensitive to the quality of available dates (Price et al., 2018). Our finer resolution highlights the importance of carefully vetting the provenance of each reported occurrence and thoroughly exploring the geological literature for such fossil occurrences—critical for any study of biochronology (Price et al., 2018) as well as selecting fossil calibrations for molecular clock dating (Parham et al., 2012).

Possible causes for the extinction of Otodus megalodon

Determination of the timing of the extinction of Otodus megalodon is a necessary step in identifying potential causal factors contributing to its demise (Pimiento & Clements, 2014; Pimiento et al., 2016). Although testing various hypotheses in a quantitative manner is beyond the scope of this article, some comments regarding potential biotic and physical drivers are appropriate given the revised extinction date presented herein. Abiotic drivers such as changes in climate, upwelling, currents, sea level, and paleogeography are possible determinants in the decline of the otodontid lineage (Pimiento et al., 2016). Physical events coincident with a “mid”-Pliocene extinction include: (1) a decrease in upwelling in the ENP (Barron, 1998), (2) increased seasonality of marine climates (Hall, 2002); (3) a period of climatic warming and permanent El-Niño like conditions in the equatorial Pacific (Wara, Ravelo & Delaney, 2005; Fedorov et al., 2013), (4) followed by late Pliocene global cooling (Zachos et al., 2001), (5) closure of the Panama seaway and restriction of currents and east-west dispersal among marine organisms (Collins et al., 1996; Haug et al., 2001; O’Dea et al., 2016; Jaramillo et al., 2017), and (6) stable eustatic sea level during the early Pliocene, (7) followed by eustatic sea level fall related to initial glaciation during the late Pliocene (Miller et al., 2005). Some of these changes in oceanic circulation and upwelling were regional, and therefore do not represent likely causes in the extinction of Otodus megalodon (if the extinction was indeed globally synchronous; e.g., Pimiento & Clements, 2014); however, these events may have been, in part, responsible for range fragmentation. Long term cooling following the middle Miocene Climatic Optimum (Zachos et al., 2001) may have reduced the geographic range of this species (Purdy, 1996; Dickson & Graham, 2004; but see Pimiento et al., 2016; Ferrón, 2017). However, a robust analysis of worldwide geographic distribution in Otodus megalodon found no change in the latitudinal distribution coincident with changes in global climate (Pimiento et al., 2016).

The lack of evidence for a climatic or geographic driver of Otodus megalodon extinction suggests that a biotic driver is probably responsible (Pimiento et al., 2016). Within the ENP, many “archaic” marine mammal taxa became extinct during the early Pleistocene (Gelasian stage, ∼2 Ma; Boessenecker, 2013a, 2013b), but the revised extinction of Otodus megalodon (this study) seems to have pre-dated this (∼3.6 Ma). The appearance of the modern marine mammal fauna appears to have occurred by the early Pliocene in the North Atlantic and western South Pacific (Whitmore, 1994; Fitzgerald, 2005; Boessenecker, 2013a), suggesting globally asymmetric origination of modern marine mammal genera and species (Boessenecker, 2013a), in contrast with an apparently synchronous extinction of Otodus megalodon (Pimiento & Clements, 2014; this study). Many extant genera of cetaceans first appeared during the Pliocene (Fordyce & Muizon, 2001), apparently temporally coincident with the extinction of Otodus megalodon, but with uncertain relevance. Other biotic effects have been hypothesized to have affected or been driven by Otodus megalodon. Recently described macrophagous sperm whales appear to have been diverse worldwide in the middle and late Miocene, were similar in size to Otodus megalodon, and were likely competing apex predators (Lambert et al., 2010). A high diversity of small-bodied baleen whales during the middle Miocene is implicated in supporting such an assemblage of gigantic predators (Lambert et al., 2010; Collareta et al., 2017). Similarly, Lindberg & Pyenson (2006) noted that the extinction of Otodus megalodon is roughly contemporaneous with the earliest fossil occurrences of killer whales (Orcinus) in the fossil record, and perhaps competition with killer whales during the Pliocene could have acted as a driver in the extinction of Otodus megalodon. However, the Neogene fossil record of Orcinus is limited to two occurrences: an isolated tooth from Japan (Kohno & Tomida, 1993), and the well-preserved skull and skeleton of Orcinus citoniensis from the late Pliocene of Italy (Capellini, 1883). Furthermore, Orcinus citoniensis was small in comparison to extant Orcinus orca (est. four m body length: Heyning & Dahlheim, 1988) and possessed a higher number of relatively smaller teeth and narrower rostrum (Bianucci, 1996), and was probably not an analogous macrophagous predator. Because fossils of Orcinus are not widespread during the Pliocene, claims of competition between Otodus megalodon and Orcinus are problematic. Furthermore, the decline and loss of cosmopolitan macrophagous physeteroids (Tortonian-Messinian; Lambert et al., 2010) appears to have predated the early Pliocene extinction of Otodus megalodon by several million years.

Evolutionary interactions with baleen whales have also been implicated for the Otodus lineage (Collareta et al., 2017). Lambert et al. (2010) and Lambert, Bianucci & De Muizon (2016) suggested that higher diversity of small-bodied mysticetes during the middle Miocene drove the evolution of killer sperm whales; similarly, this could have driven body size increases in Otodus megalodon. Cetacean diversity peaked in the middle Miocene and began to decrease in the late Miocene (Lambert et al., 2010; Marx & Uhen, 2010), and maximum body length amongst fossil mysticetes increased during the late Miocene and Pliocene (Lambert et al., 2010), heralding the appearance of modern giants such as Balaena, Balaenoptera, Eschrichtius, Eubalaena, and Megaptera. Despite the increase in maximum body size among mysticetes and apparently coincidental extinction of Otodus megalodon during the Pliocene, numerous small-bodied archaic mysticetes persisted into the Pliocene (Bouetel & Muizon, 2006; Whitmore & Barnes, 2008; Collareta et al., 2017) and even Pleistocene (Boessenecker, 2013a), complicating this relationship. A modal extinction date of 3.6 for Otodus megalodon pre-dates the extinction of certain dwarf mysticetes such as Balaenula (Piacenzian-Gelasian; Barnes, 1977), Herpetocetus (Calabrian-Ionian; Boessenecker, 2013b) and various dwarf balaenopterids (Deméré, 1986; Boessenecker, 2013a). Indeed, further study of rare late Pliocene marine mammals is necessary to further elucidate potential competition with Otodus megalodon, extinctions, and faunal dynamics (Pimiento et al., 2017).

Another potential biotic factor in the extinction of Otodus megalodon is the evolution of the modern great white shark, Carcharodon carcharias (Pimiento et al., 2016). It gradually evolved from the non-serrated Carcharodon hastalis during the late Miocene, transitioning first into the finely serrated Carcharodon hubbelli approximately 8–7 Ma, then evolved into the coarsely serrated C. carcharias approximately 6–5 Ma (Ehret, Hubbell & MacFadden, 2009a; Ehret et al., 2012; Boessenecker, 2011b; Long, Boessenecker & Ehret, 2014). However, in the western North Atlantic, C. carcharias is absent in the early Pliocene Sunken Meadow Member of the Yorktown Formation (Purdy et al., 2001; Ward, 2008), and in its place is C. hastalis (=I. hastalis and I. xiphodon in Purdy et al., 2001). Carcharodon carcharias instead occurs higher in the Rushmere Member of the Yorktown Formation (Müller, 1999). This suggests that the appearance of C. carcharias in the Atlantic may have been delayed relative to the Pacific. Pawellek et al. (2012) reported an earliest Pliocene fish assemblage on the Mediterranean coast of Libya that included C. carcharias and Otodus megalodon; clarifying the timing of first appearance of C. carcharias in ocean basins outside the Pacific is necessary, but beyond the scope of this study. Nevertheless, the timing of Otodus megalodon extinction appears to overlap with the final widespread global occurrence of C. carcharias in the early Pliocene. It is critical to note that a single putative tooth of C. carcharias has been reported from the middle Miocene Calvert Formation and has been identified as evidence supposedly disproving the C. hastalis–C. hubbelli–C. carcharias transition (Purdy, 1996; Gottfried & Fordyce, 2001), although Ehret et al. (2012) indicated this specimen is a misidentified juvenile Otodus megalodon tooth.

The development of serrations in Carcharodon hubbelli suggests a refined ability to prey upon warm-blooded prey relative to other large lamnid and carcharhinid sharks (Frazzetta, 1988; Ehret, Hubbell & MacFadden, 2009a; Ehret, MacFadden & Salas-Gismondi, 2009b; Ehret et al., 2012). Perhaps trophic competition with the newly evolved C. carcharias contributed to the extinction of Otodus megalodon, in which adult C. carcharias would have been in the same size range and likely would have competed with juvenile Otodus megalodon. Owing to its global scope, the first appearance of modern C. carcharias during the early Pliocene is a likely candidate for the driver behind the extinction of Otodus megalodon. Further investigations regarding body size trends in the Otodus and Carcharodon lineages, the C. hastalis–C. hubbelli–C. carcharias anagenetic lineage in the Pacific basin and elsewhere (Fig. 11), and the timing of C. carcharias first appearances and Otodus megalodon last appearances in the Atlantic and other ocean basins are necessary to evaluating these hypotheses of extinction drivers of Otodus megalodon.

Conclusions

Fossil teeth of Otodus megalodon have been reported from Miocene, Pliocene, and Pleistocene aged strata in California (USA) and Baja California (Mexico). Critical examination of Pleistocene specimens and their stratigraphic context clearly indicate that they are reworked, have poor provenance, or the specimens are missing (or some combination thereof), making evaluation impossible. Specimens of late Pliocene age, such as those from the Niguel Formation, also appear to be reworked from older strata. Early Pliocene specimens from the Capistrano Formation, Imperial Formation, lowermost San Diego Formation, upper San Mateo Formation, and Tirabuzón Formation appear to represent the youngest autochthonous (or parautochthonous) records, whereas numerous Otodus megalodon records of middle and late Miocene age have been reported. OLE analysis of a revised global dataset of Otodus megalodon occurrences strongly suggests that Otodus megalodon was extinct by the end of the early Pliocene (3.6 Ma), in remarkable concordance with our qualitative result based on the record from the ENP. Extinction of Otodus megalodon at 3.6 Ma appears to pre-date Pliocene-Pleistocene faunal turnover of marine mammals, and the extinction of Otodus megalodon may instead be related to late Miocene-Pliocene range fragmentation, declining numbers of small-bodied mysticete whales, and the evolution of modern Carcharodon carcharias. This study further dispels publicly held opinions that Otodus megalodon may still be extant, and demonstrates that Otodus megalodon did not survive to the end of the Pliocene. OLE results of Otodus megalodon extinction generated from our revised dataset highlight the importance of critically evaluating the locality provenance and stratigraphic control of individual fossil occurrences. Testing these revised hypotheses by the reporting of additional unpublished records of Otodus megalodon or clarifying the age of poorly constrained fossil occurrences may result in further shifts and refinements to the modal, maximum, and minimum extinction dates but absolutely requires careful assessment of geochronology and provenance.

Supplemental Information

Supplemental Information 1 Appendices. Stratigraphic justifications for Otodus megalodon occurrences used in the OLE analysis and Comments on Otodus megalodon occurrences excluded from the OLE analysis.

Click here for additional data file.

Supplemental Information 2 Code used for OLE analysis of Otodus megalodon extinction.

Code to calculate the extinction of Otodus megalodon using Optimal Linear Estimation (OLE).

Click here for additional data file.

This study benefited from discussions with J. Ashby, M. Balk, M. DeJong, T.A. Deméré, J. Duran, R.E. Fordyce, A. Gale, J. Geisler, M.D. Gottfried, S. Mansfield, F.A. Perry, C. Pimiento, and K. Shimada. We thank the following, who expedited access to collections under their care: L.G. Barnes, J. Bryant, T.A. Deméré, J. El Adli, M. Goodwin, P. Holroyd, S. McLeod, F.A. Perry, K. Randall, and V. Rhue. Thanks to S. McLeod, V. Rhue, and J. Velez-Juarbe for curatorial assistance. We are grateful for the careful comments of C.L. Powell II, K. Shimada, an anonymous reviewer and the editor K. De Baets, whose detailed comments improved the quality of an earlier draft of this study.

Institutional Abbreviations

CAS California Academy of Sciences, San Francisco, California, USA

LACM Natural History Museum of Los Angeles County, Los Angeles, California, USA

RMM Riverside Municipal Museum, Riverside, California, USA

SDNHM San Diego Natural History Museum, San Diego, California, USA

UCMP University of California Museum of Paleontology, Berkeley, California, USA.

Additional Information and Declarations

Competing Interests

Author Contributions

Data Availability

The authors declare that they have no competing interests.

Robert W. Boessenecker conceived and designed the experiments, performed the experiments, analyzed the data, contributed reagents/materials/analysis tools, prepared figures and/or tables, authored or reviewed drafts of the paper, approved the final draft.

Dana J. Ehret conceived and designed the experiments, performed the experiments, analyzed the data, contributed reagents/materials/analysis tools, authored or reviewed drafts of the paper, approved the final draft.

Douglas J. Long conceived and designed the experiments, performed the experiments, analyzed the data, contributed reagents/materials/analysis tools, authored or reviewed drafts of the paper, approved the final draft.

Morgan Churchill performed the experiments, analyzed the data, contributed reagents/materials/analysis tools, authored or reviewed drafts of the paper, approved the final draft.

Evan Martin analyzed the data, contributed reagents/materials/analysis tools, approved the final draft.

Sarah J. Boessenecker analyzed the data, contributed reagents/materials/analysis tools, prepared figures and/or tables, authored or reviewed drafts of the paper, approved the final draft.

The following information was supplied regarding data availability:

The raw data/code is included in the article (Table 1, Table 2, and Appendices). All specimens examined during this study are permanently curated at the following museums: CAS, California Academy of Sciences, San Francisco, California, USA; LACM, Natural History Museum of Los Angeles County, Los Angeles, California, USA; RMM, Riverside Municipal Museum, Riverside, California, USA; SDNHM, San Diego Natural History Museum, San Diego, California, USA; UCMP, University of California Museum of Paleontology, Berkeley, California, USA.

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
