# Peer review of "The Early Pliocene extinction of the mega-toothed shark Otodus megalodon: a view from the eastern North Pacific"

_PeerJ, doi:10.7717/peerj.6088_

## Round 0.1 · original submission · Major Revisions

Your revision of data and new finds constraining the last occurrences data of Megalodon teeth in the North Pacific is highly welcome (by all reviewers – even reviewer 3). I was a bit surprised by the rampant reworking of Megalodon teeth and a mechanism to explain it would be highly welcome – although I guess teeth are quite resistant anyway particular when they are larger (still I would mind seeing it stressed more in the text). From the way it is currently written particularly its title, parts of the introduction, some of the interpretations as well as the conclusions, I expected additional quantitative analyses to constrain true extinction ages– hence my suggestion of major revisions. However, when restructuring or rewriting parts to focus on the last confident fossil occurrences of Megalodon in California without making broadsweeping conclusions about the global extinction – it could also work and I would consider this minor revisions. I have no issue with speculation in the discussion (if it is identified as such and/or said how it could be corroborated), but you focus here mainly on occurrences in California, which is a very important, but just an initial step in re-evaluating its extinction. This should be explicitly stated in the text. As I see it relevance, I would love to see this study published either way, but I just had some crucial issues to address:

Local extinction: you study the disappareance and reworking of Megalodon teeth in various sections. If the teeth disappear in the all sections in Lower Pliocene, this could support an Mid-Pliocene local extinction, but its relevance can only be deduced when a quantitative evaluation of the uncertainty on this esimate is possible. Last Occurrence (and First Occurrence) dates are often taken at face value, but the last fossil of a taxon is a biased estimate of its true time of extinction (Wang and Marshall 2016 for a review).

Global extinction: if all last occurrences in various regions worldwide fall into the Early Pliocene, this could indicate a global extinction – however some worldwide events like sea-level changes could also affect these estimates – if appropriate facies are eroded and suitable material reworked. Furthermore, to really assess one would have to do similar studies in other regions to evaluate the local disappearance as you state here and/or also evaluate how it compared with models assuming a simultaneous extinction. As you are able to state how “wrong” certain dates are in previous studies – although this can currently not be assessed without one-by-one comparisons by the reader, you could actually use it to more reliably quantitatively evaluate the extinction of Megalodon. I guess it would be best to analyze it yourself, as sooner or later somebody else might use it to just do it.

So if you want to make a strong case for a mid-Pliocene extinction, you need to do additional analyses; if this is outside the focus of the paper, please restructure and stress how this could be used to constrain the local and regional extinction ages more confidently.

Data availability: you revise dates of fossil occurrences particularly in the North Pacific, but also globally. It would be appropriate to actually make those revised dates available as one-by-one comparisons in a supplementary table to make your claims scienfifically reproducible – currently the reader cannot possible evaluate this.

Figure 10: I would be helpful by actually adding range of allochthonous age as grey bars or stippled lines to make visible how rampant misinterpretations or reworking could have been. It would even be ok to make entire localities (or at least you did not trust) also grey or stippled in other localities to reveal how much work in revising them would be necessary.

References: you cite most but not all relevant references – in some cases you only mention crucial studies in the discussion (for example the ones discussing extinction like Collareta et al. 2017; another one is wrongly cited as Pimiento and Balk 2016 in the reference list: Pimiento et al. 2016) which would be relevant to discuss in the introduction (see annotated pdf).

We would like to see this published (see also comments by reviewers) and put quite some effort in our suggestions, so we would greatly appreciate it if you would take them into consideration.


Suggested references:

Collareta, A., Lambert, O., Landini, W., Di Celma, C., Malinverno, E., Varas-Malca, R., ... & Bianucci, G. (2017). Did the giant extinct shark Carcharocles megalodon target small prey? Bite marks on marine mammal remains from the late Miocene of Peru. Palaeogeography, Palaeoclimatology, Palaeoecology, 469, 84-91.

Pimiento, C., MacFadden, B. J., Clements, C. F., Varela, S., Jaramillo, C., Velez‐Juarbe, J., & Silliman, B. R. (2016). Geographical distribution patterns of Carcharocles megalodon over time reveal clues about extinction mechanisms. Journal of biogeography, 43(8), 1645-1655.

Wang, S. C., & Marshall, C. R. (2016). Estimating times of extinction in the fossil record. Biology letters, 12(4), 20150989.

·

Basic reporting

See attached PDF.

Experimental design

No comment.

Validity of the findings

See attached PDF.

Additional comments

I suggest the following changes. 1) Three different types of stages are used and each should be documented in every case; 2) upper, lower, middle, and late, early, middle are not capitalized unless they are formally defined as one of those terms. In the case of this manuscript that would only apply to Late or Upper Pleistocene (see http://www.stratigraphy.org/index.php/ics-chart-timescale); 3) Plio-Pleistocene is not an age, use Pliocene-Pleistocene; 4) Formation names used outside the type area and not directly correlated lithologically should be referred to in quote or ‘of authors’ (i.e., “Monterey” Formation or Monterey Formation of authors) 5) In figure 1 use a lower case e in early; 6) In figure 9 the Palos Verdes Sand does not occur around Newport Bay and these sediments should be referred to as unnamed Pleistocene sediment; 7) In figure 10 it might be nice to show a bar where the the authors believe Otodus megalodon went extinct.; 8) In table1 Ages should be listed as ICS Stages as that is what is mentioned in nearly all of them.

In addition one reference to Otodus megalodon was missed. That is by GD. Hanna (1926, Calif. Acad. Sci. Proc., 4th ser., vol. 14, No. 18, pp. 434–435). It should be included and is likely early Pliocene in age.

·

Basic reporting

The authors do a comprehensive review of the occurrence of Otodus megalodon from California and Baja California. Their literature is quite extensive and cited papers are relevant, and the entire paper is written in a clear, straight forward manner. The total number of figures may be reduced by reducing sizes of some of the imaged teeth and combining two or more figures together. Also, parts of each figure are labeled “A”, “B”, “C” etc. in capital letters, whereas the figure captions give “a”, “b”, “c” etc. in lower case, that should be standardized. All small editorial suggestions can be found in my two-part reviewed mark-up copy in PDF, and the authors must do a thorough job revising the manuscript accordingly. My biggest concern is Table 1, where the author states that their sample size is a total of 145 teeth in Line 104 but Table 1 lists only 33 specimens. The table should be comprehensive and the sample size for each rock unit must also match up with the list of specimens in Lines 147-159. In addition, each list of specimens must be cross-checked; for example, Line 486 says “14 partial teeth” but I count only 12 specimens in the same sentence. Although not critical, the authors may consider adding the following paper regarding the estimated body length for O. megalodon:
Shimada, K. 2003 (date of imprint 2002). The relationship between the tooth size and total body length in the white shark, Carcharodon carcharias (Lamniformes: Lamnidae). Journal of Fossil Research, 35(2):28–33.

Experimental design

The paper deals with the iconic ‘Megalodon’ shark and consideration about its demise; thus, it should be of general interest and is very appropriate for PeerJ. Although their analysis is rather qualitative, their methods and procedures are well described that their interpretations can be checked by others (i.e., sufficient to replicate, provided that Table 1 can list all 145 specimens they examined).

Validity of the findings

Their arguments against previous work cited as well as their findings and propositions are scientifically sound and valid

Additional comments

Although they focus on the occurrences of Otodus megalodon from California and Baja California, the authors do a good job discussing and incorporating the 'last occurrences' of the species on a global scale. I would like to see this article published in PeerJ so that I can use some of the data and discussions presented in the paper for my own work.

Reviewer 3 ·

Basic reporting

-This paper claims that the cosmopolitan Megalodon became extinct in the early Pliocene based on the last records of the species in California, without using any of the appropriate analytical methods. Accordingly, the authors base their conclusions on the assumption that the age of the last record of a cosmopolitan species in a particular region provides a reliable estimate for the date of its global extinction. Extensive literature across disciplines proves that this assumption is erroneous (i.e. species can go extinct after they were last recorded) and that quantitative methods that take into consideration the declining probability of detection are necessary to make robust extinction time estimates.
-Based on their flawed logic, the authors strongly criticize an earlier estimate of the timing of Megalodon’s extinction that does make use of global sampling and modern statistical methods. Their criticism, however, is unprofessional and ambiguous as they do not explicitly say where the other authors are wrong in their age assignments (and only present percentages of data they consider need revision). See general comments to authors for details.
-The authors do not *use* of the most recent references on the species, although they cite them (see general comments below).
-This paper is also overly long and includes many unnecessary tables and figures that add nothing to the story. It is also poorly written in parts.

I strongly recommend against publication of this article as it lacks of scientific merit.

Experimental design

-This article is not original as this question has been studied previously based on more robust analyses.
-The authors claim that there is a knowledge gap on the extinction of Megalodon, ignoring the most recent works that have been done (there are at least 5 papers published on this). While it is true that there is still a lot to be said about the extinction of this species, the authors don't recognize previous efforts and substantial findings, and they don't offer anything new or relevant.
-This work is also impossible to replicate as it is based on a qualitative and subjective observations.

Validity of the findings

- The conclusions are not linked to original research.
-Speculation is presented as main interpretation.

Additional comments

-This paper suffers from many issues that makes it unsuitable for publication. These include sloppy writing (species names not in italic, parentheses missing, incomplete sentences, etc.), exaggerated statements (e.g. that the extinction of Megalodon is understudied), outdated references (e.g. for the maximum body size of Megalodon), wrong assumptions (see basic reporting), lack of replicability (see experimental design) of their results, among others.
-This paper stands on the basis of a criticism to an early work, which is fair enough, but their criticisms are unfounded. It is also very clear that the authors fail to understand the analytical methods used in Pimiento and Clements (see comment on that paper below).
-The authors further over-interpret their qualitative work, using it to draw all kinds of conclusions about the causes of the global extinction of Megalodon, but only on the basis of their fieldwork in California and disregarding previous research. Why not simply write a paper about the last records of Megalodon in California?

On Pimento and Clements:
When thinking about the extinction of Megalodon, the first obvious step is to look at the work of Pimiento and Clements. Because this is an open access work that include the R code and data, I was willing to re run their analyses using the “correct” dates from Boessenecker et al. to see if a whole new paper with a revised date of extinction was that necessary. However, this was not possible because all criticisms to the dating of Pimiento and Clements (2014) are superficial and ambiguous. For instance, the authors main criticisms are that 1) “rather than use numerical dates from the literature, much of their dataset (88% of data consists of dates artificially stretched to fit stage ‘bins’”, that 2) “Many Piacenzian stage occurrences in New Zealand, Australia, and Europe are based on outdated stratigraphic determinations, and 3) “in many other cases (n=15, 34% of the dataset), poorly dated specimens dated to “Pliocene” are given an age of 5.3-2.6 Ma despite lacking concrete minimum dates, perhaps artificially inflating the number of true Piacenzian-age occurrences”.
First, which dates exactly are wrong (providing percentages does not help)? How do they know the dates have been stretched (I re-read the paper in question, including the supplement that provides an extensive explanation on the ages, and I see no evidence of this)? And even if they have, how is this a problem? Note that Pimento and Clements bootstrap the timing of each record from a uniform distribution between its upper and lower age. So, if the authors are aware of a new date for one of these records, if it fits between the range assigned, it still is analytically correct, except that it will reduce the uncertainty.

If these authors feel so strongly to revise the time of extinction of Megalodon, and to show that it was actually early Pliocene and not the late Pliocene, I suggest that they:
1. Make a better effort to understand the analytical approach presented in the earlier study.
2. Read the literature on estimating extinction times in the fossil record so that they understand the difference between that and LADs.
3. Re run the analyses presented in the earlier study with the new, revised dates.
They could perhaps submit a response to the PlosOne paper showing their new results.

Alternatively, since the paper does have some valuable components (e.g. the examination of the Californian sequences, valuable stratigraphic evaluation, description of specimens, etc.), the authors could write a manuscript on the last records of Megalodon in California and make inferences on local extinctions. Perhaps their analysis could elucidate a local extinction in California in the early Pliocene, before the final, global extinction in the late Pliocene... Such a study would be of great interest and could be used to refine the time of the global extinction of Megalodon, if necessary.

---

## Round 0.2 · Minor Revisions

I apologize for the delay in my response, but I wanted to wait for the response of Reviewer 3. I feel the manuscript has become clearer and more easier to follow. Also the conclusions are more in line with the crucial new data you provide. I also greatly appreciate the reanalysis using OLE. I would like to see it published, but I would still like to see some additional minor, but crucial points addressed before publication:

1) OLE: I greatly appreciate the re-analysis of the data using OLE, but you need to report the basic parameters you used (even if those are similar to those mentioned by Pimiento and Clements 2014). Crucial is also how much re-sampling/simulations you have performed. According to the summary in Figure 10, your results suggest an earlier extinction, but do no necessarily improve resolution/precision as the age range you have is larger than the one in Pimiento and Clements (2014; see also comments by reviewer). This makes logical sense as it relates with the fact that you have excluded a lot of data. Also you exclude subepoch data, but OLE should be able to deal with this to some degree (e.g., it would at least be interesting to see the differences). To be able to understand the main results, you would not only need to summarize the results as you did in Text-Fig. 10, but also plot your simulations (see Fig. 1 of Pimiento and Clements 2014) in the main text, mention the used parameters in the method section and ideally also provide the code you used in the supplementary material.

2) Introduction: Some relevant references you mention in the discussion should be also mentioned in the introduction (see comments in annotated pdf). This include a major reference (Pimiento and Balk 2015) discussing the body size of O. megalodon you wrongly cited in the previous version (and is now omitted entirely from the Introduction) as well as an additional reference discussing extinction of Megalodon (Pimiento et al. 2016). Interestingly, Pimiento and Balk (2015) even estimates a larger size of Otodus megalodon then the one you mention (Gottfried et al. 1996). Even if you disagree with this study, it should at least be mentioned why.

3) Discussion of extinction mechanisms: I agree that it appropriate to repeat the discussion of extinction mechanism and how changes in age might affect interpretations (contra the reviewer). However, I feel that additional discoveries and revision of age in different areas (particularly those currently excluded) might still shift those age and I would be appropriate to highlight this even more at the end of this paragraph.

4) Re-evaluation on previous work: You re-evaluate the data on the stratigraphy and reworking of O. megalodon material. This makes it appropriate to re-analyze the data in Pimiento and Clements (2014). However, the critique to Pimiento and Clements (2014) is duplicated several times – clearly spelling out potential issues one time in the results would be sufficient and less aggressive. Furthermore, these differences seem to be a more general issue in the O. megalodon literature, you cannot really blame solely the authors of this work. Furthermore, these authors at least used methods which should at least be partially able to deal with these issues from a statistical point of view (see comments by reviewer). The reviewer also remarked that Pimiento and Clements (2014) do not only store their data in the PBDB, but also justify their data in the supplementary material - make sure these do not contradict some of the assessment just based on the PBDB dataset.

Please address, the suggestions made in the annotated pdf and the reviewer, in addition to these points.

Looking forward to the revised version of your work.

Reviewer 3 ·

Basic reporting

Writing: There are still some errors.
Literature references and background: The authors still ignore the work on one particular author in the introduction.
Professional article structure, figs, tables: There is no figure of the analytical output.

Experimental design

Original primary research: The core of this work is not original, as the questions addressed here have been studied somewhere else before.
Research fills an identified knowledge gap: It does provide an adjustment of the extinction time, however, it was not possible to assess the analyses as the methodology of statistical analyses is not explained, if anything, their data treatment seems to be inappropriate.
Methods described with sufficient detail & information to replicate: No.

Validity of the findings

Conclusion are well stated, linked to original research question & limited to supporting results: No for the section on the causes of extinction.

Additional comments

Despite the new analyses, in which the authors adjust the time of extinction of Megalodon to between 4.14 and 3.5, as opposed to 3.5 and 2.6Ma, the authors continue 1) disregarding previous works on this species, particularly those of one author [I cannot think of a reason to do such a thing, unless they have personal, non-scientific reasons, which I think is rather unprofessional]; 2) implicitly assuming that last occurrences and time of extinction can be used interchangeably; and 3) misinterpreting the work of Pimiento and Clements (e.g. by stating that their ages were stretched to fit a time bin, which is not the case, and concluding that such practice artificially expands age ranges, which is not the case either).

There is no way to evaluate the analytical methods of this paper because they are not explained. However, given the information provided, the new analyses were not performed correctly, as the authors seem to continue not understanding the methods used in Pimiento and Clements. For instance, they excluded a number of records without subepochal age control, which is unnecessary given that all ranges are/should be re-sampled 10,000 times, bootstrapping the timing of each record from a uniform distribution between its upper and lower age to account for uncertainty. It is not possible to see if they preformed any re-sampling, as no details on the methods is provided. Most striking, they don’t include a plot, or figure on the results of their analyses.

Abstract
I suggest to be rewritten.
-It is good scientific practice to acknowledge previous works on the subject, especially if the explicitly address the question you aim to answer. “The timing of the extinction of O. megalodon is thought to be Pliocene” should be replaced with “The timing of the extinction of O. megalodon *has been proposed to be between 3.5 and 2.6Ma*”.
-Remove “although reports of Pleistocene teeth fuel speculation that O. megalodon may still be extant” because it is misleading: 1) Previous works have already assessed the timing of extinction of this species, and there is no longer any speculation. 2) The Pleistocene records are still fossil and by themselves, don’t fuel any speculation on the existence of the species today. 3) The only speculation that exists is on the media.
-Why well-sampled and well-dated upper Pliocene strata is needed? Don’t assume the reader knows that the last records are important to assess time of extinction, explain.
-L36, no space after ;
-L36-37: “youngest reliable records of O. megalodon are early Pliocene, suggesting an extinction at the early-late Pliocene boundary (~3.6 Ma)”. The authors insist in regarding last occurrence as extinction time, which is not the case.
-In general, the abstract does not have a logic flow: When rewriting, consider: Start with a few sentences on a basic introduction to the field, then provide a more detailed background, then state the problem being addressed here, then provide an overview of your results, then explain what the main results means compared with previous studies and how it adds to previous knowledge, finally provide some sentences on a the general context of your results and the broader perspective.

Introduction
Again, it is good scientific practice to acknowledge previous works on the subject, especially if the explicitly address the question you aim to answer.
-L50-52: Pimiento et al. 2016 specifically addresses the geographic distribution of megalodon.
-L58-59: Pimiento and Balk 2015 estimated a max TL of 18m. If you have reasons to not believe this work, you should still acknowledge it (e.g. Otodus megalodon is estimated to have attained a body length of 16 m (Gottfried et al., 1996, *BUT SEE PIMIENTO AND BALK 2015)
-L62-68: When providing what has been done to understand this species, the authors insist on ignoring the work of Pimiento (body size: Pimiento and Balk 2015; geographic distribution: Pimiento et al. 2016). I am now convinced that the authors have reasons beyond science to exclude the work of this particular author.

Material and Methods
-L114: predating the… (?)
-L126: “instead relying (?) upon the ages cited within the paleontological papers reporting the fossil occurrences”. Misleading. See Text S1 from Pimiento and Clements and it is clear that they revised the ages: Here some examples:
--PaleoBioDB# 18548: Early Pliocene age clearly explained in text. However, when studying specimens in the collections of the Florida Museum of Natural History (FLMNH), a Hemphillian age was regarded for this locality…
--PaleoBioDB# 18577: Early Pliocene age clearly explained in text and confirmed when
studying specimens in the collections of the FLMNH.
-L127: “Readers are instructed to consult the Paleobiology Database (Pimiento and Clements, 2014:2). Misleading. Justifications are explicitly provided in text S1.
-L138-139: Poor writing
-L149: Explain what OLE does. The authors don’t seem to be concerned with this paper being so long, therefore, they should also spend some lines explaining the analytical methods. They don’t provide any detail on how analyses were performed, and therefore, this reviewer cannot evaluate it (e.g. It is not possible to assess if the authors indeed resampled their data (or how many times) or if they performed any simulations.
-There is absolutely no reason to exclude records with lack of subepoch control when using OLE and the methods of Pimiento and Clements (i.e. because uncertainty is taken into account by re-sampling the fossil data 10,000 times, bootstrapping the timing of each record from a uniform distribution between its upper and lower age).

Results
The word “corrections” should be replaced with “adjustments” throughout.
-L 559-560: The data from Pimiento and Clements is not artificially stretched to fit stage ‘bins”. It is clear from reading their paper and looking a Tables S1-2 that they used min and max ages as reported in the literature. Therefore, the bins used correspond to those ages and were not stretched to fit any stage bin. Hence, it is incorrect to state that this artificially expanded the age range. It is true that when finer resolution is available it should be used instead, but the way the authors are phrasing this here is misleading. Rephrase or remove.
-L575-580: There is no reason to exclude records from Bone Valley Formation as Morgan, 1994 confirms its age.
-587-589: Again, there is no reason to exclude records with broad ranges as the methods used in Pimiento and Clements account for time uncertainties.
-The authors say in the abstract that they used the median and in the results that they used the modal value to estimate extinction time. Which one is it? Cannot evaluate this as no details on the analyses are provided.
-L604-607: Why should we believe that those are the results you got? Where is the plot? Fig 10 does not represent a plot of the output from OLE, as it only consist on a line drawn artificially. How can the modal value 3.51 and the youngest possible extinction 3.17 Ma. These two values should be the same. There is something wrong with these analyses but impossible to judge.
-The OLE needs some simulations as presented in Pimiento and Clements. Did you perform any? How many of the simulations fall in the range proposed?
-L720-723: The authors insist to use last records and time of extinction interchangeably. That is simply wrong and there is plenty of literature explaining why.

Discussion
The tone of the critiques to Pimiento and Clements continue to be unfair. As in Wang and Marshall (2015), the lack of resolution of fossil data is problem of paleontology, and authors using such data should not be blamed for it. After all, they do what they can with the data available, and use the best methods available to cope with the problems of the fossil record. Consequently, poor resolution in the results of Pimiento and Clements should not be interpreted as bad practice, as the authors here implicitly state.
-L825-829: It is not true that this study has a better resolution than that of Pimiento and Clements, their estimated time of extinction is 3.5-2.6, whereas this one is 4.1-3.5. So the vetting used by the authors did not improve the resolution at all.
-The section on possible causes for the extinction of Otodus megalodon seems to be a repetition of what has been published elsewhere and seems out of place here. At least, there is no correspondence between this and the data they used.
-L842-844: The most recent papers on the timing of the closure of the central American seaway are not considered here (Farris et al. 2011; Montes et al. 2012a; Montes et al. 2012b; Bacon et al. 2015a, b; Montes et al. 2015; Jaramillo et al. 2017).

---

## Round 0.3 · accepted · Accept

Thank you for integrating/addressing my final suggestions. Particularly the description of the details of the OLE analysis and the addition of a figure showing its results make it easier to understand the results and follow the discussion. I also greatly appreciate that you addressed some minor reference inconsistencies and worked on the tone of the manuscript. In science, we might now always agree on approaches and interpretations. Often, this also pushes forward our knowledge as different parties do additional analyses to further test and back up hypotheses. However, both authors and reviewers, should strive to remain polite and objective. This is not always easy as it can be quite emotional when we do not agree with particular analyses or if research counters ideas/data we have worked for long time, but this is how science works. Looking forward to seeing this manuscript published.

#